# A Near-Optimal Algorithm for Debiasing Trained Machine Learning Models

**Ibrahim Alabdulmohsin**
Google Research, Brain Team
Zürich, Switzerland
ibomohsin@google.com

**Mario Lucic**
Google Research, Brain Team
Zürich, Switzerland
lucic@google.com

## Abstract

We present a scalable post-processing algorithm for debiasing trained models, including deep neural networks (DNNs), which we prove to be near-optimal by bounding its excess Bayes risk. We empirically validate its advantages on standard benchmark datasets across both classical algorithms as well as modern DNN architectures and demonstrate that it outperforms previous post-processing methods while performing on par with in-processing. In addition, we show that the proposed algorithm is particularly effective for models trained at scale where post-processing is a natural and practical choice.

## 1  Introduction

**Background.**  Machine learning is increasingly applied to critical decisions which can have a lasting impact on individual lives, such as for credit lending [Bruckner, 2018], medical applications [Deo, 2015], and criminal justice [Brennan et al., 2009]. Consequently, it is imperative to understand and improve the degree of bias of such automated decision-making.

Unfortunately, despite the fact that bias (or "fairness") is a central concept in our society today, it is difficult to define it in precise terms. In fact, as people perceive ethical matters differently depending on a plethora of factors including geographical location or culture [Awad et al., 2018], no universally-agreed upon definition for bias exists. Moreover, bias may depend on the application and might even be ignored in favor of accuracy when the stakes are high, such as in medical diagnosis [Kleinberg et al., 2016]. As such, it is not surprising that several measures of bias have been introduced, such as statistical parity [Dwork et al., 2012, Zafar et al., 2017a], equality of opportunity [Hardt et al., 2016], and equalized odds [Hardt et al., 2016, Kleinberg et al., 2016], and these are not generally mutually compatible [Chouldechova, 2017, Kleinberg et al., 2016].

Let $\mathcal{X}$ be an instance space and let $\mathcal{Y} = \{0, 1\}$ be the target set in a binary classification problem. In the fair classification setting, we may further assume the existence of a sensitive attribute $s : \mathcal{X} \to \{1, \dots, K\}$, where $s(x) = k$ if and only if $x \in X_k$ for some total partition $\mathcal{X} = \cup_k X_k$. For example, $\mathcal{X}$ might correspond to the set of job applicants while $s$ indicates their sex. Then, a commonly used criterion for fairness is to require similar mean outcomes across the sensitive attribute (a.k.a. statistical parity) [Dwork et al., 2012, Zafar et al., 2017a, Mehrabi et al., 2019]:

**Definition 1** (Statistical Parity). *Let $\mathcal{X}$ be an instance space and $\mathcal{X} = \cup_k X_k$ be a total partition of $\mathcal{X}$. A predictor $h : \mathcal{X} \to [0, 1]$ satisfies $\epsilon$ statistical parity across all groups $X_1, \dots, X_K$ if:*

$$\max_{k \in [K]} \mathbb{E}_{\boldsymbol{x}}[h(\boldsymbol{x}) \,|\, \boldsymbol{x} \in X_k] - \min_{k \in [K]} \mathbb{E}_{\boldsymbol{x}}[h(\boldsymbol{x}) \,|\, \boldsymbol{x} \in X_k] \leq \epsilon,$$

*where $[K]$ denotes the set $\{1, \dots, K\}$.*

Our main contribution is to derive a near-optimal recipe for debiasing models, including deep neural networks, according to Definition 1. Specifically, we formulate the task of debiasing learned models

35th Conference on Neural Information Processing Systems (NeurIPS 2021).

as a regularized optimization problem that is solved efficiently using the projected SGD method. We show how the algorithm produces thresholding rules with randomization near the thresholds, where the width of randomization is controlled by a regularization hyperparamter. We also prove that randomization near the threshold is, in general, necessary for Bayes risk consistency. In Appendix D, we show how the proposed algorithm can be modified to handle a weaker notion of bias as well. We refer to the proposed algorithm as Randomized Threshold Optimizer (RTO).

Besides the theoretical guarantees, we empirically validate RTO on benchmark datasets across both classical algorithms as well as modern DNN architectures. Our experiments demonstrate that the proposed algorithm significantly outperforms previous post-processing methods and performs competitively with in-processing (Section 5). While we focus on binary sensitive attributes in the experiments, our algorithm and its guarantees continue to hold for non-binary attributes as well.

In addition, we show that RTO is particularly effective for models trained at scale where post-processing is a natural and practical choice. Qualitatively speaking, for a fixed downstream task $D$, such as predicting facial attributes in the CelebA dataset [Liu et al., 2015], we say that the model is "trained at scale" if it is both: (1) heavily overparameterized, and (2) pretrained on large datasets before fine-tuning on the downstream task $D$. We show that the impact of debiasing models on their performance using the proposed algorithm can be improved with scale.

**Remark.**    Because "bias" is a societal concept that cannot be reduced to metrics such as statistical parity [Chouldechova, 2017, Dixon et al., 2018, Selbst et al., 2019], our conclusions do not necessarily pertain to "fairness" in its broader sense. Rather, they hold for the narrow technical definition of statistical parity. Similarly, we conduct experiments on standard benchmark datasets, such as CelebA [Liu et al., 2015] and COCO [Lin et al., 2014], which are commonly used in the literature, as a way of validating the technical claims of this paper. Our experiments are, hence, not to be interpreted as an endorsement of those visions tasks, such as predicting facial attributes.

## 2   Related Work

Algorithms for fair machine learning can be broadly classified into three groups: (1) pre-processing methods, (2) in-processing methods, and (3) post-processing methods [Zafar et al., 2019].

Preprocessing algorithms transform the data into a different representation such that any classifier trained on it will not exhibit bias. This includes methods for learning a fair representation [Zemel et al., 2013, Lum and Johndrow, 2016, Bolukbasi et al., 2016, Calmon et al., 2017, Madras et al., 2018, Kamiran and Calders, 2012], label manipulation [Kamiran and Calders, 2009], data augmentation [Dixon et al., 2018], or disentanglement [Locatello et al., 2019].

On the other hand, in-processing methods constrain the behavior of learning algorithms in order to control bias. This includes methods based on adversarial training [Zhang et al., 2018] and constraint-based classification, such as by incorporating constraints on the decision margin [Zafar et al., 2019] or features [Grgić-Hlača et al., 2018]. Agarwal et al. [2018] showed that the task of learning an unbiased classifier could be reduced to a *sequence* of cost-sensitive classification problems, which could be applied to any black-box classifier. One caveat of the latter approach is that it requires solving a linear program (LP) and retraining classifiers, such as neural networks, *many* times before convergence.

The algorithm we propose in this paper is a post-processing method, which can be justified theoretically [Corbett-Davies et al., 2017, Hardt et al., 2016, Menon and Williamson, 2018, Celis et al., 2019]. Fish et al. [2016] and Woodworth et al. [2017] fall under this category. However, the former only provides generalization guarantees without consistency results while the latter proposes a two-stage approach that requires changes to the original training algorithm. Kamiran et al. [2012] also proposes a post-processing algorithm, called Reject Option Classifier (ROC), without any theoretical guarantees. In contrast, our algorithm is Bayes consistent and does not alter the original classification method. In Celis et al. [2019] and Menon and Williamson [2018], instance-dependent thresholding rules are also learned. However, our algorithm also learns to *randomize* around the threshold (Figure 1(a)) and this randomization is *key* to our algorithm both theoretically as well as experimentally (Appendix B and Section 5). Hardt et al. [2016] learns a randomized post-processing rule but our proposed algorithm outperforms it in all of our experiments (Section 5). Also, [Wei et al., 2019] is a post-processing method but it requires solving a non-linear optimization problem (for the dual variables) via ADMM and provides guarantees for approximate fairness only.

Woodworth et al. [2017] showed that the post-processing approach can be suboptimal. Nevertheless, the latter result does not contradict the statement that our post-processing rule is near-optimal because we assume that the original classifier outputs a score (i.e. a monotone transformation of an approximation to the posterior $p(\mathbf{y} = 1 \,|\, \mathbf{x})$ such as margin or softmax output) whereas Woodworth et al. [2017] assumed that the post-processing rule had access to the binary predictions only.

We argue that the proposed algorithm has distinct advantages, particularly for deep neural networks (DNNs). First, stochastic convex optimization methods can scale well to massive amounts of data [Bottou, 2010], which is often the case in deep learning today. Second, the guarantees provided by our algorithm hold w.r.t. the *binary* predictions instead of using a proxy, such as the margin as in some previous works [Zafar et al., 2017b, 2019]. Third, unlike previous reduction methods that would require retraining a deep neural network several times until convergence [Agarwal et al., 2018], which can be prohibitively expensive, our algorithm does not require retraining. Also, post-processing can be the *only* available option, such as when using machine learning as a service with out-of-the-box predictive models or due to various other constrains in data and computation [Yang et al., 2020b].

# 3   Near-Optimal Algorithm for Statistical Parity

**Notation.**   We reserve boldface letters for random variables (e.g. $\mathbf{x}$), small letters for instances (e.g. $x$), capital letters for sets (e.g. $X$), and calligraphic typeface for universal sets (e.g. the instance space $\mathcal{X}$). Given a set $S$, $1_S(x) \in \{0, 1\}$ is its characteristic function. Also, we denote $[N] = \{1, \dots, N\}$ and $[x]^+ = \max\{0, x\}$. We reserve $\eta(x)$ for the Bayes regressor: $\eta(x) = p(\mathbf{y} = 1 \,|\, \mathbf{x} = x)$.

**Algorithm.**   Given a classifier outputting a probability score $\hat{p}(\mathbf{y} = 1 \,|\, \mathbf{x} = x)$, let $f(x) = 2\hat{p}(\mathbf{y} = 1 \,|\, \mathbf{x} = x) - 1$. We refer to $f(x) \in [-1, +1]$ as the classifier's output. Our goal is to post-process the predictions made by the classifier to control statistical parity with respect to a sensitive attribute $s : \mathcal{X} \to [K]$ according to Definition 1. To this end, instead of learning a deterministic rule, we consider *randomized* prediction rules $h(\mathbf{x})$, where $h(\mathbf{x})$ is the probability of predicting the positive class given $f(\mathbf{x})$ and $s(\mathbf{x})$. Note that we have the Markov chain: $\mathbf{x} \to (s(\mathbf{x}), f(\mathbf{x})) \to h(\mathbf{x})$.

A simple approach of achieving $\epsilon$ statistical parity is to output a constant prediction in each subpopulation, which is clearly suboptimal in general. As such, there is a tradeoff between accuracy and fairness. The approach we take in this work is to modify the original classifier such that the original predictions are matched as much as possible while satisfying the fairness constraints. Minimizing the probability of altering the binary predictions of the original classifier can be achieved by maximizing the inner product $\mathbb{E}_{\mathbf{x}}[h(\mathbf{x}) \cdot f(\mathbf{x})]$ (cf. Appendix B). However, maximizing this objective alone leads to *deterministic* thresholding rules which have a major drawback as illustrated in the following example.

**Example 1** (Randomization is necessary)**.** *Suppose that $\mathcal{X} = \{-1, 0, 1\}$ where $p(\mathbf{x} = -1) = 1/2$, $p(\mathbf{x} = 0) = 1/3$ and $p(\mathbf{x} = 1) = 1/6$. Let $\eta(-1) = 0$, $\eta(0) = 1/2$ and $\eta(1) = 1$. In addition, let $s \in \{0, 1\}$ be a sensitive attribute, where $p(s = 1|\mathbf{x} = -1) = 1/2$, $p(s = 1|\mathbf{x} = 0) = 1$, and $p(s = 1|\mathbf{x} = 1) = 0$. Then, the Bayes optimal prediction rule $h^\star(x)$ subject to statistical parity ($\epsilon = 0$) satisfies: $p(h^\star(\mathbf{x}) = 1|\mathbf{x} = -1) = 0$, $p(h^\star(\mathbf{x}) = 1|\mathbf{x} = 0) = 7/10$ and $p(h^\star(\mathbf{x}) = 1|\mathbf{x} = 1) = 1$.*

As a result, randomization close to the threshold is *necessary* in the general case to achieve Bayes risk consistency. This conclusion in Example 1 remains true with approximate fairness ($\epsilon < 12/70$). In this work, we propose to achieve this by minimizing the following *regularized* objective for some hyperparameter $\gamma > 0$:

$$(\gamma/2)\, \mathbb{E}_{\mathbf{x}}[h(\mathbf{x})^2] \; - \; \mathbb{E}_{\mathbf{x}}[h(\mathbf{x}) \cdot f(\mathbf{x})]. \tag{4}$$

We prove in Appendix A that this regularization term leads to randomization around the threshold, which is critical, both theoretically (Section 4 and Appendix B) and experimentally (Section 5). Informally, $\gamma$ controls the width of randomization as illustrated in Figure 1.

Let $\mathcal{X} = \cup_k X_k$ be a total partition of the instance space according to the sensitive attribute $s : \mathcal{X} \to [K]$. Denote a finite training sample by $\mathcal{S} = \{(x_1, y_1), \dots, (x_N, y_N)\}$ and write $S_k = \mathcal{S} \cap X_k$. For each group $S_k$, the fairness constraint in Definition 1 over the training sample can be written as:

$$\frac{1}{|S_k|} \Big| \sum_{x_i \in S_k} (h(x_i) - \rho) \Big| \; \leq \; \frac{\epsilon}{2}, \tag{5}$$

for some hyper-parameter $\rho \in [0, 1]$. Precisely, if the optimization variables $h(x_i)$ satisfy the constraint (5), then Definition 1 holds over the training sample by the triangle inequality. Conversely,

**Algorithm 1:** Pseudocode of the Randomized Threshold Optimizer (RTO).

---

**Input:** $\gamma > 0; \rho \in [0, 1]; \epsilon \geq 0; f : \mathcal{X} \rightarrow [-1, 1]; s : \mathcal{X} \rightarrow [K]$
**Output:** Prediction rule: $h_\gamma(x)$
**Training:** Initialize $(\lambda_1, \mu_1), \ldots, (\lambda_K, \mu_K)$ to zeros. Then, repeat until convergence:

1. Sample an instance $\mathbf{x} \sim p(x)$

2. Perform the updates:

$$\lambda_{s(\mathbf{x})} \leftarrow [\lambda_{s(\mathbf{x})} - \eta g_{\lambda_{s(\mathbf{x})}}]^+, \quad \mu_{s(\mathbf{x})} \leftarrow [\mu_{s(\mathbf{x})} - \eta g_{\mu_{s(\mathbf{x})}}]^+ \tag{1}$$

where:

$$g_{\lambda_{s(\mathbf{x})}} = \frac{\epsilon}{2} + \rho + \frac{\partial}{\partial \lambda_{s(\mathbf{x})}} \xi_\gamma \big( f(\mathbf{x}) - (\lambda_{s(\mathbf{x})} - \mu_{s(\mathbf{x})}) \big)$$

$$g_{\mu_{s(\mathbf{x})}} = \frac{\epsilon}{2} - \rho + \frac{\partial}{\partial \mu_{s(\mathbf{x})}} \xi_\gamma \big( f(\mathbf{x}) - (\lambda_{s(\mathbf{x})} - \mu_{s(\mathbf{x})}) \big).$$

and:

$$\xi_\gamma(w) = \frac{w^2}{2\gamma} \cdot \mathbb{I}\{0 \leq w \leq \gamma\} + \big(w - \frac{\gamma}{2}\big) \cdot \mathbb{I}\{w > \gamma\} \tag{2}$$

**Prediction:** Given an instance $x$ in the group $X_k$, predict the label $+1$ with probability $h_\gamma(x)$, where:

$$h_\gamma(x) = \big[\min\{1, (f(x) - \lambda_k + \mu_k)/\gamma\}\big]^+ \tag{3}$$

---

if Definition 1 holds, then the constraint (5) also holds where:

$$2\rho = \max_{k \in [K]} \mathbb{E}_\mathbf{x}[h(\mathbf{x}) \mid \mathbf{x} \in S_k] + \min_{k \in [K]} \mathbb{E}_\mathbf{x}[h(\mathbf{x}) \mid \mathbf{x} \in S_k].$$

Therefore, to learn the post-processing rule $h(x)$, we solve the optimization problem:

$$\min_{0 \leq h(x_i) \leq 1} \qquad \sum_{x_i \in \mathcal{S}} (\gamma/2) h(x_i)^2 - f(x_i) h(x_i)$$

$$\text{s.t.} \qquad \forall k \in [K] : \Big| \sum_{x_i \in S_k} (h(x_i) - \rho) \Big| \leq \epsilon_k, \tag{6}$$

in which $\epsilon_k = |S_k| \epsilon/2$ for all $k \in [K]$. Using Lagrange duality we show in Appendix A that solving the above optimization problem is equivalent to Algorithm 1. In Appendix D, we show that if $\epsilon = 0$, an alternative formulation can be used to minimize the same objective while satisfying the fairness constraint but *without* introducing a hyperparameter $\rho$. To reiterate, $\rho \in [0, 1]$ is tuned via a validation dataset and $\gamma > 0$ is a hyperparameter that controls randomization.

## 4 Theoretical Analysis

Our first theoretical result is to show that RTO satisfies the desired fairness guarantees.

**Theorem 1** (Correctness). *Let $h_\gamma : \mathcal{X} \rightarrow [0, 1]$ be the randomized predictor in Equation 3 learned by applying the update rules in Equation 1 on a fresh sample of size $N$ until convergence with learning rates satisfying the Robbins and Monro condition [Robbins and Monro, 1951]. Then, $h_\gamma$ satisfies $\epsilon$ statistical parity on the training sample. Moreover, with a probability of at least $1 - \delta$, the following bound on bias holds w.r.t. the underlying distribution:*

$$\max_{k \in [K]} \mathbb{E}[h(\boldsymbol{x}) \mid \boldsymbol{x} \in X_k] - \min_{k \in [K]} \mathbb{E}[h(\boldsymbol{x}) \mid \boldsymbol{x} \in X_k] \leq \epsilon + 8\sqrt{\frac{2 \log \frac{eN}{2}}{N}} + 2\sqrt{\frac{\log \frac{2K}{\delta}}{N}}. \tag{7}$$

*Proof.* The proof is in Appendix A. We make use of strong duality, which holds by Slater's condition [Boyd and Vandenberghe, 2004]. The update rules correspond to the projected SGD method on the dual problem. This establishes the guarantee on the training sample. For the underlying distribution, we bound the Rademacher complexity [Bousquet et al., 2003] of the function class $\mathcal{H}_\gamma$ of Figure 1(a) by that of 0-1 thresholding rules over $\mathbb{R}$, from which a generalization bound is derived. $\square$

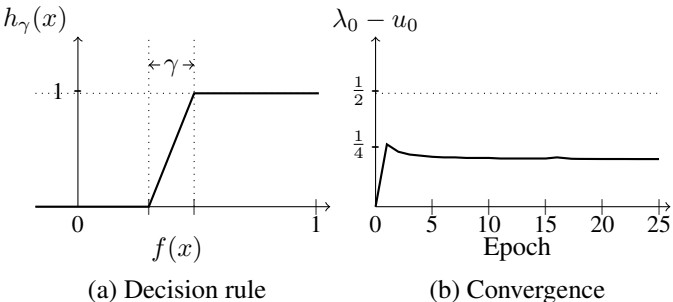

(a) Decision rule      (b) Convergence

Figure 1: (a) The learned post-processing rule $h_\gamma(x)$ in Equation 3 as a function of the classifier's score $f(x)$ over one subpopulation. Randomization is applied when $h_\gamma(x) \in (0,1)$. (b) The value of $\lambda_0 - u_0$ is plotted against the number of epochs in projected SGD applied to the random forests classifier. The classifier is trained on the Adult dataset to implement statistical parity with respect to the sex attribute (cf. Section 5). We observe fast convergence in agreement with Proposition 1.

The following guarantee shows that the randomized prediction rule converges to the Bayes optimal unbiased classifier if the original classifier is Bayes consistent.

**Theorem 2.** *Let $h^\star = \arg\min_{h \in \mathcal{H}_\epsilon} \mathbb{E}[h(\boldsymbol{x}) \neq \boldsymbol{y}]$, where $\mathcal{H}_\epsilon$ is the set of binary predictors on $\mathcal{X}$ that satisfy fairness on the training sample according to Definition 1 for $\epsilon \geq 0$. Let $h_\gamma : \mathcal{X} \to [0,1]$ be the randomized rule in Algorithm 1. If $h_\gamma$ is trained on a fresh data of size $N$, then there exists a value of $\rho \in [0,1]$ independent of $N$ such that the following holds with a probability of at least $1 - \delta$:*

$$\mathbb{E}[\mathbb{I}\{h_\gamma(\boldsymbol{x}) \neq \boldsymbol{y}\}] \leq \mathbb{E}[\mathbb{I}\{h^\star(\boldsymbol{x}) \neq \boldsymbol{y}\}] + 2\gamma + \frac{8(2 + \frac{1}{\gamma})}{N^{\frac{1}{3}}} + \mathbb{E}|2\eta(\boldsymbol{x}) - 1 - f(\boldsymbol{x})| + 4\sqrt{\frac{2K + 2\log \frac{2}{\delta}}{N}},$$

*where $\eta(x) = p(\boldsymbol{y} = 1 | \boldsymbol{x} = x)$ is the Bayes regressor and $K$ is the number of groups $X_k$.*

*Proof.* The full proof is in Appendix B. First, we show that minimizing the probability of error can be achieved by maximizing $\mathbb{E}[f(\mathbf{x}) \cdot (2\eta(\mathbf{x}) - 1)]$. We use the regularized loss instead, which is strongly convex. Using Lipschitz continuity of the decision rule when $\gamma > 0$ (cf. Figure 1(a)), and the robustness framework of Xu and Mannor [2012], we prove a generalization bound and proceed with a series of inequalities to establish the main theorem. $\qquad\square$

Thus, if the original classifier is Bayes consistent, namely $\mathbb{E}|2\eta(\mathbf{x}) - 1 - f(\mathbf{x})| \to 0$ as the sample size goes to infinity, and if $N \to \infty$, $\gamma \to 0^+$ and $\gamma N^{\frac{1}{3}} \to \infty$, then $\mathbb{E}[h_\gamma(\mathbf{x}) \neq \mathbf{y}] \xrightarrow{P} \mathbb{E}[h^\star(\mathbf{x}) \neq \mathbf{y}]$. Hence, Algorithm 1 converges to the *optimal* prediction rule subject to the fairness constraints.

**Convergence Rate.** As we show in Appendix A, the update rules in Equation 1 perform a projected stochastic gradient descent on the following optimization problem:

$$\min_{\mu, \lambda \geq 0} F = \mathbb{E}_{\mathbf{x}}\Big[\epsilon\left(\lambda_{s(\mathbf{x})} + \mu_{s(\mathbf{x})}\right) + \rho\left(\lambda_{s(\mathbf{x})} - \mu_{s(\mathbf{x})}\right) + \xi_\gamma(f(\mathbf{x}) - (\lambda_{s(\mathbf{x})} - \mu_{s(\mathbf{x})}))\Big], \qquad (8)$$

where $\xi_\gamma$ is given by Equation 2. The following proposition shows that the post-processing rule can be efficiently computed. In practice, we observe fast convergence as demonstrated in Figure 1(b).

**Proposition 1.** *Let $\mu^{(0)} = \lambda^{(0)} = 0$ and write $\mu^{(t)}, \lambda^{(t)} \in \mathbb{R}^K$ for the value of the optimization variables after $t$ updates defined in Equation 1 for some fixed learning rate $\alpha_t = \alpha$. Let $\bar{\mu} = (1/T)\sum_{t=1}^T \mu^{(t)}(x)$ and $\bar{\lambda} = (1/T)\sum_{t=1}^T \lambda^{(t)}(x)$. Then,*

$$\mathbb{E}[\bar{F}] - F^\star \leq (1 + \rho + \epsilon)^2 \alpha + \frac{||\mu^\star||_2^2 + ||\lambda^\star||_2^2}{2T\alpha}, \qquad (9)$$

*where $\bar{F} : \mathbb{R}^K \times \mathbb{R}^K \to \mathbb{R}$ is the objective function in (8) using the averaged solution $\bar{\mu}$ and $\bar{\lambda}$ while $F^\star$ is its optimal value. In particular, $\mathbb{E}[\bar{F}] - F^\star = \mathcal{O}(\sqrt{K/T})$ when $\alpha = \mathcal{O}(\sqrt{K/T})$.*

The proof of Proposition 1 is in Appendix C. As shown in Figure 1(a), the hyperparameter $\gamma$ controls the width of randomization around the thresholds. A large value of $\gamma$ may reduce the accuracy of the classifier. On the other hand, $\gamma$ cannot be zero because randomization around the threshold is, in general, necessary for Bayes risk consistency as shown earlier in Example 1.

# 5 Experiments

**Baselines and Experimental Setup.**    We compare against three post-processing methods: (1) the algorithm of Hardt et al. [2016] (2) the shift inference method, first introduced in Saerens et al. [2002] and used more recently in Wang et al. [2020b], and (3) the Reject Option Classifier (ROC) [Kamiran et al., 2012]. We also include the reduction approach of Agarwal et al. [2018] to compare the performance against in-processing rules. We briefly review each of these methods next.

The post-processing method of Hardt et al. [2016] is a randomized post-processing rule. It was originally developed for equalized odds and equality of opportunity. Nevertheless, it can be modified to accommodate other criteria, such as statistical parity [Agarwal et al., 2018, Dudik et al., 2020].

The shift inference rule, on the other hand, is a post-hoc correction that views bias as a shift in distribution, hence the name. It is based on the identity $r(\mathbf{y}|\mathbf{s}, \mathbf{x}) \propto q(\mathbf{y}|\mathbf{s}, \mathbf{x}) \cdot r(\mathbf{y}, \mathbf{s})/q(\mathbf{y}, \mathbf{s})$, which holds for any two distributions $r$ and $q$ on the product space of labels $\mathbf{y}$, sensitive attributes $\mathbf{s}$, and instances $\mathbf{x}$ if they share the same marginal $r(\mathbf{x}) = q(\mathbf{x})$ [Wang et al., 2020b]. By equating, $q(\mathbf{y}|\mathbf{s}, \mathbf{x})$ with the classifier's output based on the biased distribution and $r(\mathbf{y}|\mathbf{s}, \mathbf{x})$ with the unbiased classifier, the predictions of the classifier $q(\mathbf{y}|\mathbf{s}, \mathbf{x})$ can be post-hoc corrected for bias by multiplying its probability score with the ratio $p(\mathbf{y})p(\mathbf{s})/p(\mathbf{y}, \mathbf{s})$.

The reject option classifier (ROC) proposed by Kamiran et al. [2012] is a deterministic thresholding rule. It enumerates all possible values of some tunable parameter $\theta$ up to a given precision, where $\theta = 0$ corresponds to the original classifier. Candidate thresholds are then tested on the data.

Finally, the reduction approach of Agarwal et al. [2018] is an in-processing method that can be applied to black-box classifiers but it requires retraining the model several times. More precisely, let $h$ be a hypothesis in the space $\mathcal{H}$ and $M$ be a matrix, Agarwal et al. [2018] showed that minimizing the error of $h$ subject to constraints of the form $M\mu(h) \leq c$, where $\mu(h)$ is a vector of conditional moments on $h$ of a particular form, can be reduced (with some relaxation) to a sequence of cost-sensitive classification tasks for which many algorithms can be employed.

We use the implementations of Hardt et al. [2016] and Agarwal et al. [2018] in the FairLearn software package [Dudik et al., 2020]. The training data used for the post-processing methods is always a fresh sample, i.e. different from the data used to train the original classifiers. Specifically, we split the data that was not used in the original classifier into three subsets of equal size: (1) training data for the post-processing rules, (2) validation for hyperparameter selection, and (3) test data. The value of the hyper-parameter $\theta$ of the ROC algorithm is chosen in the grid $\{0.01, 0.02, \ldots, 1.0\}$. In the proposed algorithm, the parameter $\gamma$ is chosen in the grid $\{0.01, 0.02, 0.05, 0.1, 0.2\}$ while $\rho$ is chosen in the gird $\mathbb{E}[\mathbf{y}] \pm \{0, 0.05, 0.1\}$. All hyper-parameters are selected based on a separate validation dataset. For the in-processing approach, we used the Exponentiated Gradient method as proposed by Agarwal et al. [2018] with its default settings in the FairLearn package (e.g. max iterations of 50).

**Tabular Data.**    We evaluate performance on two real-world datasets, namely the Adult income dataset [Kohavi, 1996] and the Default of Credit Card Clients (DCCC) dataset [Yeh and Lien, 2009], both from the UCI Machine Learning Repository [Blake and Merz, 1998]. The Adult dataset contains 48,842 records with 14 attributes each and the goal is to predict if the income of an individual exceeds $50K per year. The DCCC dataset contains 30,000 records with 24 attributes, and the goal is to predict if a client will default on their credit card payment. We set sex as a sensitive attribute. In DCCC, we introduce bias to the training set to study the case in which bias shows up in the training data only (e.g. due to the data curation process) but the test data remains unbiased (cf. [Torralba and Efros, 2011] and [de Vries et al., 2019] who discuss similar observations in common benchmark datasets). Specifically, if $s(\mathbf{x}) = y(\mathbf{x})$ we keep the instance and otherwise drop it with probability 0.5.

We train four classifiers: (1) random forests with depth 10, (2) $k$-NN with $k = 10$, (3) a two-layer neural network with 128 hidden nodes, and (4) logistic regression whose parameter $C$ is fine-tuned from a grid of values in a logarithmic scale between $10^{-4}$ and $10^4$ using 10-fold cross validation. The learning rate in our algorithm is fixed to $10^{-1}(K/T)^{1/2}$, where $T$ is the number of steps, and $\epsilon = 0$.

Table 1 (Top and Middle) shows the bias on *test* data after applying each post-processing method. The column marked as "original" corresponds to the original classifier without alteration. As shown in the table, the shift-inference method does not succeed at controlling statistical parity while ROC can fail when the original classifier's output is concentrated on a few points because it does not randomize.

Table 1: A comparison of four post-processing methods and the reduction approach of Agarwal et al. [2018] on 3 datasets. The classifiers are random forests (RF), $k$-NN, MLP, logisitic regression (LR), ResNet50 trained from scratch (R50/S), ResNet50 pretrained on ImageNet (R50/I), MobileNet trained from scratch (MN/S) and MobileNet pretrained on ImageNet (MN/I). Values in **bold** correspond to cases where debiasing **fails**. ROC may fail in $k$-NN and in neural networks because debiasing them can require randomization. Original bias in the dataset is provided in the leftmost column.

| | | **Bias** | | | | | |
|---|---|---|---|---|---|---|---|
| Dataset | Classifier | *Original* | RTO | Hardt, 2016 | Shift Inference | ROC | Reduction |
| ADULT | RF | *.38* | .01 | .01 | **.16** | .02 | .01 |
| (Bias = .19) | $k$NN | *.24* | .02 | .01 | **.08** | **.08** | .01 |
| | MLP | *.29* | .01 | .02 | **.10** | .02 | .01 |
| | LR | *.39* | .01 | .02 | **.10** | .01 | .01 |
| DCCC | RF | *.07* | .01 | .01 | **.09** | .02 | .01 |
| (Bias = .21) | $k$NN | *.10* | .01 | .01 | **.18** | .02 | .01 |
| | MLP | *.13* | .01 | .01 | **.12** | .02 | .01 |
| | LR | *.12* | .01 | .01 | **.13** | .01 | .01 |
| CELEBA | R50/S | *.43* | .01 | .01 | **.38** | **.08** | ⋆ |
| (Bias = .33) | R50/I | *.40* | 0.02 | .01 | **.35** | **.15** | ⋆ |
| | MN/S | *.35* | .01 | .01 | **.24** | .01 | ⋆ |
| | MN/I | *.38* | .002 | .002 | **.34** | **.10** | ⋆ |

**CelebA Dataset.** Our second set of experiments builds on the task of predicting the "attractiveness" attribute in the CelebA dataset [Liu et al., 2015]. We reiterate that we do not endorse the usage of vision models for such tasks, and that we report these results because they exhibit sex-related bias. CelebA contains 202,599 images of celebrities annotated with 40 binary attributes, including sex. We use two standard architectures: ResNet50 [He et al., 2016] and MobileNet [Howard et al., 2017], trained from scratch or pretrained on ImageNet ILSVRC2012 [Deng et al., 2009]. We resize images to $224 \times 224$ and train with a fixed learning rate of 0.001 until the validation error converges. We present the bias results in Table 1 (bottom). We observe that randomization is indeed necessary: ROC and Shift Inference both fail at debiasing the neural networks because they do not learn to randomize when most scores produced by neural networks are concentrated around the set $\{-1, +1\}$.

**Impact on Test Accuracy.** As shown in Table 2, the proposed algorithm has a much lower impact on the test accuracy compared to Hardt et al. [2016] and even improves the test accuracy in DCCC because bias was introduced in DCCC to the training data only as discussed earlier. The tradeoff curves between accuracy and bias for both the proposed algorithm and Hardt et al. [2016] are shown in Figure 2 (LEFT). Also, for a comparison with in-processing rules, we observe that the post-processing algorithm performs competitively with the reduction approach of Agarwal et al. [2018].

**Impact of Scale.** Models trained at scale transfer better and enjoy improved out-of-distribution robustness [Djolonga et al., 2021]. As these models are now often used in practice, we assess to which extent can these models be debiased while retaining high accuracy. We conduct 768 experiments on 16 deep neural networks architectures, pretrained on either ILSVRC2012, ImageNet-21k (a superset of ILSVRC2012 that contains 21k classes [Deng et al., 2009]), or JFT-300M (a proprietary dataset with 300M examples and 18k classes [Sun et al., 2017]). The 16 architectures are listed in Appendix E and include MobileNet [Howard et al., 2017], DenseNet [Huang et al., 2017], Big Trasnfer (BiT) models [Kolesnikov et al., 2020], and NASNetMobile [Zoph et al., 2018]. The classification tasks contain seven attribute prediction tasks in CelebA [Liu et al., 2015] as well as five classification tasks based on the COCO dataset [Lin et al., 2014]. We describe how the tasks were selected in Appendix E. The sensitive attribute is always sex in our experiments and all classification tasks are binary. Unless explicitly stated, we use $\epsilon = 0$. Moreover, in the COCO dataset, we follow the procedure of [Wang et al., 2020a] in inferring the sensitive attribute based on the image caption: we use images that contain either the word "woman" or the word "man" in their captions but not both.

In every task, we build a linear classifier on top of the pretrained features. Inspired by the HyperRule in [Kolesnikov et al., 2020], we train for 50 epochs with an initial learning rate of 0.003, which is dropped by factor of 10 after 20, 30, and 40 epochs. All images are resized to $224 \times 224$. For augmentation, we use random horizontal flipping and cropping, where we increase the dimension of the image to $248 \times 248$ before cropping an image of size $224 \times 224$ at random.

Table 2: A comparison of the test accuracy of the proposed algorithm against the algorithms of Hardt et al. [2016] and the reduction approach of Agarwal et al. [2018]. Both Shift Inference and ROC failed at debiasing all models (Table 1) so they are excluded from the comparison here.

| | | **Test Accuracy** | | | |
|---|---|---|---|---|---|
| | | *Original* | RTO | Hardt, 2016 | Reduction |
| | RF | $85.7 \pm .1\%$ | $84.4 \pm .1\%$ | $81.0 \pm .2\%$ | $83.9 \pm .1\%$ |
| ADULT | $k$NN | $86.8 \pm .1\%$ | $81.3 \pm .2\%$ | $78.7 \pm .2\%$ | $80.2 \pm .1\%$ |
| | MLP | $85.5 \pm .2\%$ | $83.5 \pm .3\%$ | $79.7 \pm .2\%$ | $83.5 \pm .1\%$ |
| | LR | $84.9 \pm .2\%$ | $83.0 \pm .1\%$ | $79.4 \pm .2\%$ | $83.3 \pm .2\%$ |
| | RF | $81.2 \pm .2\%$ | $81.8 \pm .1\%$ | $80.6 \pm .2\%$ | $81.4 \pm .3\%$ |
| DCCC | $k$NN | $79.6 \pm .2\%$ | $80.4 \pm .2\%$ | $78.7 \pm .1\%$ | $79.5 \pm .1\%$ |
| | MLP | $80.5 \pm .1\%$ | $81.3 \pm .2\%$ | $78.8 \pm .2\%$ | $81.3 \pm .2\%$ |
| | LR | $80.6 \pm .2\%$ | $81.7 \pm .1\%$ | $78.3 \pm .1\%$ | $80.5 \pm .3\%$ |
| | R-S | $77.8\%$ | $71.3\%$ | $65.9\%$ | $\star$ |
| CELEBA | R-I | $79.7\%$ | $71.7\%$ | $67.5\%$ | $\star$ |
| | M-S | $76.9\%$ | $71.8\%$ | $66.4\%$ | $\star$ |
| | M-I | $79.3\%$ | $72.8\%$ | $67.5\%$ | $\star$ |

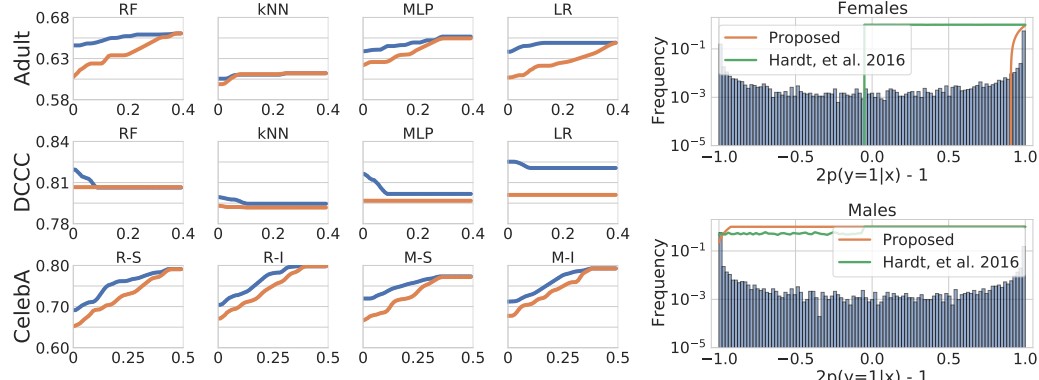

Figure 2: LEFT: The tradeoff curves are displayed for each classification problem, where blue curves are for the proposed RTO algorithm and amber curves are for Hardt et al. [2016]. The $x$-axis is bias (Definition 1) while the $y$-axis is test accuracy. RIGHT: The distribution of the scores produced by ResNet50 trained from scratch are shown for both subpopulations. The curves correspond to $p(\mathbf{y} = 1|\mathbf{x})$ of Hardt et al. [2016] and the proposed algorithm when $\gamma = 0.1$ and $\rho = \mathbb{E}[\mathbf{y}]$.

**Scaling up the Model Size.** First, we examine the impact of over-parameterization in pretrained models on the effectiveness of the proposed post-processing algorithm. We fix the upstream dataset to ILSVRC2012 (8 models in total, cf. Appendix E) and aggregate the test error rates across tasks by placing them on a common scale using *soft ranking*. Specifically, we rescale all error rates in a given task linearly, so that the best error achieved is zero while the largest error is one. After that, we average the performance of each model across all tasks. Aggregated results are given in Figure 3. The impact of the proposed algorithm on test errors improves by scaling up the size of pretrained models.

**Scaling up the Data.** Second, we look into the impact of the size of the upstream data. We take the four BiT models ResNet50x1, ResNet50x3, ResNet101x1 and ResNet101x3, each is pretrained on either ILSVRC2012, ImageNet-21k, or JFT-300K [Kolesnikov et al., 2020]. For each model and every downstream task, we rank the upstream datasets according to the test error on the downstream task and report the average ranking. Figure 4 shows that pretraining each model on JFT-300M yields the best test accuracy when it is debiased using the proposed algorithm. To ensure that the improvement is not solely due to the data collection process, we pretrain ResNet50 on subsets of ImageNet-21k before fine-tuning on the 12 downstream tasks. Figure 5 shows, again, that the impact of the proposed post-processing rule on test errors improve when pretraining on large datasets.

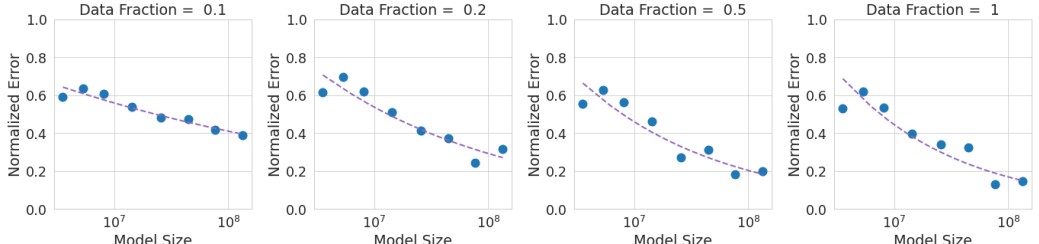

Figure 3: Aggregated performance of debiased DNN models pretrained on ILSVRC2012 across 12 classification tasks in CelebA and COCO (see Appendix E). The $x$-axis is the number of model parameters while the $y$-axis is the aggregated error rate across all tasks after normalization (see Section 5). Figures from left to right use 10%, 20%, 50%, & 100% of downstream data, respectively.

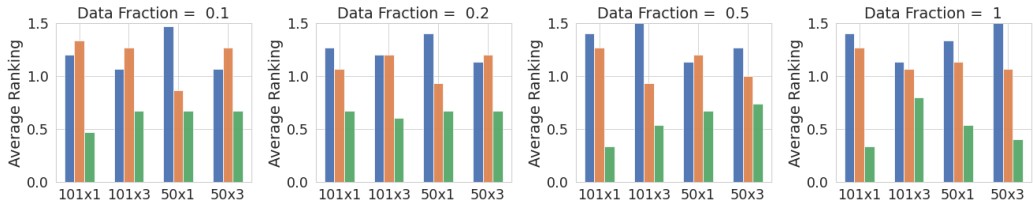

Figure 4: Aggregated performance of debiased Big Transfer (BiT) models pretrained on ILSVRC2012 (blue), ImageNet-21k (orange), or JFT-300M (green). The y-axis is the average ranking of each upstream dataset (lower is better) according to the test error rate on each of the 12 downstream classification tasks in Appendix E. Figures from left to right use 10%, 20%, 50%, & 100% of downstream data, respectively. In all models, pretraining on JFT-300K yields the best performance.

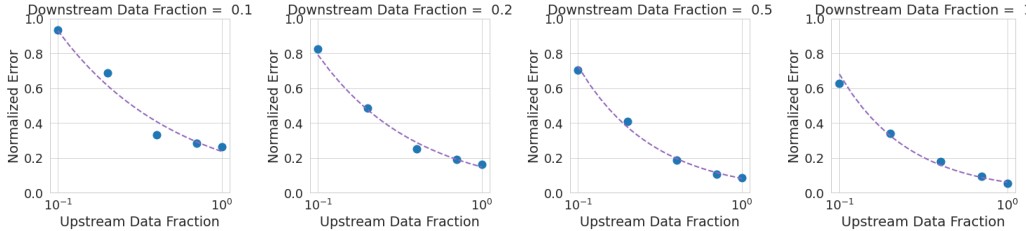

Figure 5: Aggregated performance of debiasing ResNet50 when pretrained on subsets of ImageNet-21k across 12 classification tasks in CelebA and COCO (see Appendix E). The $x$-axis is the fraction of ImageNet-21k used during pretraining while the $y$-axis follows the approach in Figure 3. Figures from left to right use 10%, 20%, 50%, & 100% of downstream data, respectively. The impact of the proposed post-processing algorithm on test errors improves when pretraining on large datasets.

## 6   Conclusion

The post-processing approach in fair classification enjoys many advantages. It can be applied to any classification algorithm and does not require retraining. In addition, it is sometimes the *only* option available, such as when using machine learning as a service with out-of-the-box predictive models [Obermeyer et al., 2019] or due to other constraints in data and computation [Yang et al., 2020a].

In this paper, we propose a near-optimal scalable post-processing algorithm for debiasing trained models according to statistical parity. In addition to its strong theoretical guarantees, we show that it outperforms previous post-processing methods on standard benchmark datasets across classical and modern machine learning models, and performs favorably with even in-processing methods. Finally, we show that the algorithm is particularly effective for models trained at scale, in which heavily overparameterized models are pretrained on large datasets before fine-tuning on the downstream task.

## Acknowledgement

The authors are grateful to Lucas Dixon, Daniel Keysers, Ben Zevenbergen, Philippe Gervais, Mike Mozer and Olivier Bousquet for the valuable comments and discussions.

## Funding Disclosure

This work was performed at and funded by Google. The authors declare that there is no conflict of interest.

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
