# A Proof of Theorem 1

## A.1 Proof of Correctness on the Training Sample

Here we repeat the setup from Section 3 for completeness.

Suppose we have a binary classifier on the instance space $\mathcal{X}$. Let $f : \mathcal{X} \to [-1, +1]$ be its scores as described in Section 3. We would like to construct an algorithm for post-processing the predictions made by that classifier such that we control the bias with respect to a set of pairwise disjoint groups $X_1, \ldots, X_K \subseteq \mathcal{X}$ according to Definition 1. We assume that the output of the classifier $f : \mathcal{X} \to [-1, +1]$ is an estimate to $2\eta(x) - 1$, where $\eta(x) = p(\mathbf{y} = 1|\mathbf{x} = x)$ is the Bayes regressor. This is not a strong assumption because many algorithms can be calibrated to provide probability scores [Platt et al., 1999, Guo et al., 2017] so the assumption is valid. We consider randomized rules $h(x)$ that post-process the original classifier's output $f(x)$ according to the sensitive attribute $s(x)$. Because randomization is sometimes necessary as demonstrated in Example 1, $h(x)$ is the probability of predicting the positive class when the instance is $x \in \mathcal{X}$.

In a finite training sample of size $N$, which we will denote by $\mathcal{S} = \{(x_1, y_1), \ldots, (x_N, y_N)\}$, let $S_k = S \cap X_k$. For each group $X_k \subseteq \mathcal{S}$, the fairness constraint in Definition 1 over the training sample can be written as:

$$\frac{1}{|S_k|} \Big| \sum_{x_i \in S_k} (h(x_i) - \rho) \Big| \leq \frac{\epsilon}{2},$$

for some hyper-parameter $\rho > 0$. Precisely, if the optimization variables $h(x_i)$ satisfy the constraint (5), then Definition 1 holds by the triangle inequality. Conversely, if Definition 1 holds, then the constraint (5) also holds where:

$$2\rho = \max_{k \in [K]} \mathbb{E}_{\mathbf{x}}[h(\mathbf{x}) \,|\, \mathbf{x} \in X_k] + \min_{k \in [K]} \mathbb{E}_{\mathbf{x}}[h(\mathbf{x}) \,|\, \mathbf{x} \in X_k].$$

To learn $h$, we propose solving the following *regularized* optimization problem:

$$\min_{0 \leq h(x) \leq 1} \qquad \sum_{x_i \in \mathcal{S}} (\gamma/2)\, h(x_i)^2 - f(x_i)\, h(x_i)$$

$$\text{s.t.} \qquad \forall k \in [K] : \Big| \sum_{x_i \in S_k} (h(x_i) - \rho) \Big| \leq \epsilon_k, \qquad (10)$$

where $\gamma > 0$ is a regularization parameter and $\epsilon_k = |S_k|\, \epsilon/2$ for all $k \in [K]$.

Because the groups $X_k$ are pairwise disjoint, the optimization problem in (10) decomposes into $K$ separate suboptimization problems, one for each group $X_k$. Each sub-optimization problem can be written in the following general form, where $q_i = h(x_i)$:

$$\min_{0 \leq q_i \leq 1} \qquad \sum_{i=1}^{M} \frac{\gamma}{2} q_i^2 - f(x_i) q_i$$

$$\text{s.t.} \qquad \sum_{i=1}^{M} (z_i q_i - b) \leq \epsilon', \qquad -\sum_{i=1}^{M} (z_i q_i - b) \leq \epsilon',$$

for some values of $M \in \mathbb{N}$ and $z_i, b \in \mathbb{R}$, where $\epsilon' = M\epsilon/2$.

**Note:** We introduce new symbols $M, z_i$ and $b$ to keep the subsequent analysis general. For (10), in particular, $M$ would correspond to the size of the group $S_k$, $z_i = 1$, and $b = \rho$. Later in Appendix D, we show that another criterion of bias falls into this general form so the same analysis applies over there as well.

The Lagrangian of the convex optimization problem is:

$$L(q, \alpha, \beta, \lambda, \mu) = \sum_{i=1}^{M} \left( \frac{\gamma}{2} q_i^2 - f(x_i) q_i \right)$$

$$+ \lambda \Big( \sum_{i=1}^{M} (z_i q_i - b) - \epsilon' \Big) - \mu \Big( \sum_{i=1}^{M} (z_i q_i - b) + \epsilon' \Big) + \sum_{i=1}^{M} \alpha_i (q_i - 1) - \sum_{i=1}^{M} \beta_i q_i.$$

Taking the derivative w.r.t. $q_i$ gives us:

$$q_i = \frac{1}{\gamma}\Big(f(x_i) - (\lambda - \mu)z_i - \alpha_i + \beta_i\Big).$$

Plugging this back, the dual problem becomes:

$$\min_{q,\lambda,\mu,\alpha,\beta} \quad \sum_{i=1}^{M}\Big(\frac{\gamma}{2}q_i^2 + b(\lambda - \mu)\Big) + (\lambda + \mu)\epsilon' + \sum_{i=1}^{M}\alpha_i$$

$$\text{s.t.} \quad q_i = \frac{1}{\gamma}\Big(f(x_i) - (\lambda - \mu)z_i - \alpha_i + \beta_i\Big)$$

$$\lambda, \mu, \alpha_i, \beta_i \geq 0.$$

Next, we eliminate variables. By eliminating $\beta_i$, we have:

$$\min_{q,\lambda,\mu,\alpha,\beta} \quad \sum_{i=1}^{M}\Big(\frac{\gamma}{2}q_i^2 + b(\lambda - \mu)\Big) + (\lambda + \mu)\epsilon' + \sum_{i=1}^{M}\alpha_i$$

$$\text{s.t.} \quad q_i - \frac{1}{\gamma}\Big(f(x_i) - (\lambda - \mu)z_i - \alpha_i\Big) \geq 0$$

$$\lambda, \mu, \alpha_i \geq 0.$$

Equivalently:

$$\min_{q,\lambda,\mu,\alpha,\beta} \quad \sum_{i=1}^{M}\Big(\frac{\gamma}{2}q_i^2 + b(\lambda - \mu)\Big) + (\lambda + \mu)\epsilon' + \sum_{i=1}^{M}\alpha_i$$

$$\text{s.t.} \quad \alpha_i \geq f(x_i) - \gamma q_i - (\lambda - \mu)z_i$$

$$\lambda, \mu, \alpha_i \geq 0.$$

Next, we eliminate $\alpha_i$ to obtain:

$$\min_{q,\lambda,\mu} \quad \sum_{i=1}^{M}\Big(\frac{\gamma}{2}q_i^2 + b(\lambda - \mu)\Big) + (\lambda + \mu)\epsilon' + \sum_{i=1}^{M}\big[f(x_i) - \gamma q_i - (\lambda - \mu)z_i\big]^{+}$$

$$s.t. \quad \lambda, \mu \geq 0.$$

Finally, we eliminate the $q_i$ variables. For a given optimal $\mu$ and $\lambda$, it is straightforward to observe that the minimizer $q^\star$ to $\gamma/2q^2 + [w - \gamma q]^{+}$ must lie in the set $\{0, w/\gamma, 1\}$. In particular, if $w/\gamma \leq 0$, then $q^\star = 0$. If $w/\gamma \geq 1$, then $q^\star = 1$. Note here that we make use of the fact that $\gamma > 0$.

So, the optimal value of $q^\star$ to $\gamma/2q^2 + [w - \gamma q]^{+}$ is:

$$\xi_\gamma(w) = \begin{cases} 0 & \frac{w}{\gamma} \leq 0 \\ \frac{w^2}{2\gamma} & 0 \leq \frac{w}{\gamma} \leq 1 \\ w - \frac{\gamma}{2} & \frac{w}{\gamma} \geq 1 \end{cases}$$

From this, the optimization problem reduces to:

$$\min_{\lambda,\mu \geq 0} \quad \sum_{i=1}^{M}\Big(b(\lambda - \mu) + \epsilon'(\lambda + \mu) + \xi_\gamma(f(x_i) - (\lambda - \mu)z_i)\Big). \tag{11}$$

This is a differentiable objective function and can be solved quickly using the projected gradient descent method [Boyd and Mutapcic, 2008]. The projection step here is taking the positive parts of $\lambda$ and $\mu$. This leads to the update rules in Algorithm 1.

Finally, given $\lambda$ and $\mu$, the solution of $q_i$ is a minimizer to;

$$\frac{\gamma}{2}q_i^2 + \big[f(x_i) - \gamma q_i - (\lambda - \mu)z_i\big]^{+}.$$

This solution is given by Equation (3). So, we have a ramp function. In the proposed algorithm, we have $z_i = 1$ and $b = \rho$ for all examples. The Robbins and Monro conditions on the learning rate schedule guarantee convergence to the optimal solution [Robbins and Monro, 1951]. This proves Theorem 1.

## A.2 Generalization to Test Data

The previous section establishes the correctness of the proposed algorithm on the training sample. Therefore, upon termination, one has for every subpopulation $X_k$ with $S_k \doteq \mathcal{S} \cap X_k$:

$$\frac{1}{S_k} \Big| \sum_{x_i \in S_k} h(x_i) - \rho \Big| \leq \frac{\epsilon}{2}.$$

This guarantee holds on the training sample. However, since the decision rule $h(x)$ is a ramp function of the form shown in Figure 1(a), which is learned according to a fresh training sample of size $N$, the bias guarantee generalizes to test data as well as shown next.

First, let $\hat{\mathcal{R}}(\mathcal{H}_\gamma)$ be the conditional Rademacher complexity of the hypothesis class that comprises of functions $h_\gamma : \mathbb{R} \to [0, 1]$ of the form:

$$h_\gamma(z) = \begin{cases} 0 & z \leq b \\ (1/\gamma)(z - b) & b < z < b + \gamma \\ 1 & z \geq b + \gamma \end{cases},$$

which are depicted in Figure 1(a). We show that $\hat{\mathcal{R}}(\mathcal{H}_\gamma)$ is bounded by the conditional Rademacher complexity of 0-1 thresholding rules over the real line $\mathbb{R}$. By definition, for a fixed training sample $\{z_1, \ldots, z_N\}$ [Bousquet et al., 2003]:

$$\hat{\mathcal{R}}(\mathcal{H}_\gamma) = \mathbb{E}_\sigma \sup_{h \in \mathcal{H}_\gamma} \frac{1}{N} \sum_{i=1}^{N} \sigma_i h(z_i). \tag{12}$$

Given fixed instances of the Rademacher random variables $\sigma_i \in \{-1, +1\}$, let $h_\sigma^\star(z)$ be the function that achieves the supremum inside the expectation. We note that if $\sum_i \sigma_i \mathbb{I}\{0 < h_\sigma^\star(z_i) < 1\} > 0$, then the 0-1 thresholding rule $h'(z) = \mathbb{I}\{z \geq b\}$ satisfies:

$$\frac{1}{N} \sum_{i=1}^{N} \sigma_i h'(z_i) \geq \frac{1}{N} \sum_{i=1}^{N} \sigma_i h_\sigma^\star(z_i).$$

Conversely, if $\sum_i \sigma_i \mathbb{I}\{0 < h_\sigma^\star(z_i) < 1\} \leq 0$, then the 0-1 thresholding rule $h'(z) = \mathbb{I}\{z \geq b + \gamma\}$ satisfies the above inequality. This shows that the conditional Rademacher complexity of 0-1 thresholding rules is, at least, as large as the conditional Rademacher complexity of $\mathcal{H}_\epsilon$. However, by classical counting results that relate the Rademacher complexity to the VC dimension, we conclude:

$$\mathcal{R}_n(\mathcal{H}_\gamma) \leq 2\sqrt{\frac{2 \log \frac{en}{2}}{N}},$$

because the VC dimension of the 0-1 thresholding rules over the real line is 2. Thus, for any fixed subpopulation $X_k$, one has with a probability of at least $1 - \delta$:

$$\big| \mathbb{E}[h(\mathbf{x}) \,|\, \mathbf{x} \in X_k] - \rho \big| \leq \frac{\epsilon}{2} + 2\mathcal{R}_n(\mathcal{H}_\gamma) + \sqrt{\frac{\log \frac{2}{\delta}}{N}} \leq \frac{\epsilon}{2} + 4\sqrt{\frac{2 \log \frac{en}{2}}{N}} + \sqrt{\frac{\log \frac{2}{\delta}}{N}}.$$

In addition, by the union bound, we have with a probability of at least $1 - \delta$, the following inequalities all hold simultaneously:

$$\forall k \in [K] \ : \ \big| \mathbb{E}[h(\mathbf{x}) \,|\, \mathbf{x} \in X_k] - \rho \big| \leq \frac{\epsilon}{2} + 4\sqrt{\frac{2 \log \frac{en}{2}}{N}} + \sqrt{\frac{\log \frac{2K}{\delta}}{N}}.$$

Hence, with a probability of at least $1 - \delta$:

$$\max_{k \in [K]} \mathbb{E}[h(\mathbf{x} \,|\, \mathbf{x} \in X_k] - \min_{k \in [K]} \mathbb{E}[h(\mathbf{x} \,|\, \mathbf{x} \in X_k] \leq \epsilon + 8\sqrt{\frac{2 \log \frac{en}{2}}{N}} + 2\sqrt{\frac{\log \frac{2K}{\delta}}{N}}. \tag{13}$$

# B  Proof of Theorem 2

## B.1  Optimal Unbiased Predictors

We begin by proving the following result, which can be of independent interest.

**Theorem 3.** *Let $f^\star = \arg\min_{f:\mathcal{X}\to\{0,1\}} \mathbb{E}[\mathbb{I}\{f(\boldsymbol{x}) \neq \boldsymbol{y}\}]$ be the Bayes optimal decision rule subject to group-wise affine constraints of the form $\mathbb{E}[w_k(\boldsymbol{x}) \cdot f(\boldsymbol{x}) \,|\, \boldsymbol{x} \in X_k] = b_k$ for some fixed partition $\mathcal{X} = \cup_k X_k$. If $w_k : \mathcal{X} \to \mathbb{R}$ and $b_k \in \mathbb{R}$ are such that there exists a constant $c \in (0,1)$ in which $p(f(x) = 1) = c$ will satisfy all the affine constraints, then $f^\star$ satisfies $p(f^\star(x) = 1) = \mathbb{I}\{\eta(x) > t_k\} + \tau_k \,\mathbb{I}\{\eta(x) = t_k\}$, where $\eta(x) = p(\boldsymbol{y} = 1|\boldsymbol{x} = x)$ is the Bayes regressor, $t_k \in [0,1]$ is a threshold specific to the group $X_k \subseteq \mathcal{X}$, and $\tau_k \in [0,1]$.*

*Proof.* Minimizing the expected misclassification error rate of a classifier $f$ is equivalent to maximizing:

$$\mathbb{E}[f(\mathbf{x}) \cdot \mathbf{y} + (1 - f(\mathbf{x})) \cdot (1 - \mathbf{y})] = \mathbb{E}\Big[\mathbb{E}_{\mathbf{x}}[f(\mathbf{x}) \cdot \mathbf{y} + (1 - f(\mathbf{x})) \cdot (1 - \mathbf{y})] \,\big|\, \mathbf{x}\Big]$$
$$= \mathbb{E}\Big[\mathbb{E}_{\mathbf{x}}[f(\mathbf{x}) \cdot (2\eta(\mathbf{x}) - 1)] \,\big|\, \mathbf{x}\Big] + \mathbb{E}[1 - \eta(\mathbf{x})].$$

Hence, selecting $f$ that minimizes the misclassification error rate is equivalent to maximizing:

$$\mathbb{E}[f(\mathbf{x}) \cdot (2\eta(\mathbf{x}) - 1)]. \tag{14}$$

Instead of maximizing this directly, we consider the regularized form first. Writing $g(x) = 2\eta(x) - 1$, the optimization problem is:

$$\min_{0 \leq f(x) \leq 1} \quad (\gamma/2)\mathbb{E}[f(\mathbf{x})^2] - \mathbb{E}[f(\mathbf{x}) \cdot g(\mathbf{x})]$$
$$\text{s.t.} \quad \mathbb{E}[w(\mathbf{x}) \cdot f(\mathbf{x})] = b$$

Here, we focused on one subset $X_k$ because the optimization problem decomposes into $K$ separate optimization problems, one for each $X_k$. If there exists a constant $c \in (0,1)$ such that $f(x) = c$ satisfies all the equality constraints, then Slater's condition holds so strong duality holds [Boyd and Vandenberghe, 2004]. Note that in the case of fair classification, this is always the case because having a fixed $f(x) = c$ yields a predictor that is independent of the instances so the fairness constraints are satisfied.

The Lagrangian is:

$$(\gamma/2)\mathbb{E}[f(\mathbf{x})^2] - \mathbb{E}[f(\mathbf{x}) \cdot g(\mathbf{x})] + \mu(\mathbb{E}[w(\mathbf{x}) \cdot f(\mathbf{x})] - b) + \mathbb{E}[\alpha(\mathbf{x})(f(\mathbf{x}) - 1)] - \mathbb{E}[\beta(\mathbf{x})f(\mathbf{x})],$$

where $\alpha(x), \beta(x) \geq 0$ and $\mu \in \mathbb{R}$ are the dual variables.

Taking the derivative w.r.t. the optimization variable $f(x)$ yields:

$$\gamma f(x) = g(x) - \mu\, w(x) - \alpha(x) + \beta(x). \tag{15}$$

Therefore, the dual problem becomes:

$$\max_{\alpha(x),\beta(x) \geq 0} \quad -(2\gamma)^{-1}\,\mathbb{E}[(g(\mathbf{x}) - \mu\, w(\mathbf{x}) - \alpha(\mathbf{x}) + \beta(\mathbf{x}))^2] - b\mu - \mathbb{E}[\alpha(\mathbf{x})].$$

We use the substitution in Equation (15) to rewrite it as:

$$\min_{\alpha(x),\beta(x) \geq 0} (\gamma/2)\,\mathbb{E}[f(\mathbf{x})^2] + b\mu + \mathbb{E}[\alpha(\mathbf{x})]$$
$$\text{s.t.}\forall x \in \mathcal{X} : \gamma f(x) = g(x) - \mu\, w(x) - \alpha(x) + \beta(x).$$

Next, we eliminate the multiplier $\beta(x)$ by replacing the equality constraint with an inequality:

$$\min_{\alpha(x) \geq 0} (\gamma/2)\,\mathbb{E}[f(\mathbf{x})^2] + b\mu + \mathbb{E}[\alpha(\mathbf{x})]$$
$$\text{s.t.}\forall x \in \mathcal{X} : g(x) - \gamma f(x) - \mu\, w(x) - \alpha(x) \leq 0.$$

Finally, since $\alpha(x) \geq 0$ and $\alpha(x) \geq g(x) - \gamma f(x) - \mu w(x)$, the optimal solution is the minimizer to:

$$\min_{f:\mathcal{X}\to\mathbb{R}} (\gamma/2)\mathbb{E}[f(\mathbf{x})^2] + b\mu + \mathbb{E}[\max\{0,\, g(\mathbf{x}) - \gamma f(\mathbf{x}) - \mu w(\mathbf{x})\}].$$

Next, let $\mu^\star$ be the optimal solution of the dual variable $\mu$. Then, the optimization problem over $f$ decomposes into separate problems, one for each $x \in \mathcal{X}$. We have:

$$f(x) = \arg\min_{\tau\in\mathbb{R}}\left\{(\gamma/2)\tau^2 + [g(x) - \gamma\tau - \mu^\star w(x)]^+\right\}.$$

Using the same argument in Appendix A, we deduce that $f(x)$ is of the form:

$$f(x) = \begin{cases} 0, & g(x) - \mu^\star w(x) \leq 0 \\ 1 & g(x) - \mu^\star w(x) \geq \gamma \\ (1/\gamma)\,(g(x) - \mu^\star w(x)) & \text{otherwise} \end{cases}$$

Finally, the statement of the theorem holds by taking the limit as $\gamma \to 0^+$. $\qquad\square$

## B.2   Excess Risk Bound

In this section, we write $\mathcal{D}$ to denote the underlying probability distribution and write $\mathcal{S}$ to denote the uniform distribution over the training sample (a.k.a. empirical distribution).

The parameter $\rho$ stated in the theorem is given by:

$$\rho = (1/2)\Big(\max_{k\in[K]}\mathbb{E}_{\mathbf{x}}[h^\star(\mathbf{x}) \mid \mathbf{x} \in X_k] + \min_{k\in[K]}\mathbb{E}_{\mathbf{x}}[h^\star(\mathbf{x}) \mid \mathbf{x} \in X_k]\Big).$$

Note that, by definition, the optimal classifier $h^\star$ that satisfies $\epsilon$ statistical parity also satisfies the constraint in (6) with this choice of $\rho$. Hence, with this choice of $\rho$, $h^\star$ remains optimal among all possible classifiers.

Observe that the decision rule depends on $x$ only via $f(x) \in [-1, +1]$. Hence, we write $\mathbf{z} = f(\mathbf{x})$. Since the thresholds are learned based on a fresh sample of data, the random variables $\mathbf{z}_i$ are i.i.d. In light of Equation 14, we would like to minimize the expectation of the loss $l(h_\gamma, \mathbf{x}) = -f(\mathbf{x})\cdot h_\gamma(\mathbf{x}) = -\mathbf{z}\cdot q(\mathbf{z}) \doteq \zeta(\mathbf{z})$ for some function $q : [-1, +1] \to [0, 1]$ of the form shown in 1(a). Note that $\zeta$ is $2(1 + 1/\gamma)$-Lipschitz continuous within the same group and sensitive class. This is because the thresholds are always in the interval $[-1 - \gamma, 1 + \gamma]$; otherwise moving beyond this interval would not change the decision rule.

Let $\mathbf{h}_\gamma$ be the decision rule learned by the algorithm. Using Corollary 5 in [Xu and Mannor, 2012], we conclude that with a probability of at least $1 - \delta$:

$$\left|\mathbb{E}_\mathcal{D}[l(\mathbf{h}_\gamma, \mathbf{x})] - \mathbb{E}_\mathcal{S}[l(\mathbf{h}_\gamma, \mathbf{x})]\right| \leq \inf_{R\geq 1}\left\{\Big(\frac{4}{R}\big(1 + \frac{1}{\gamma}\big)\Big) + 2\sqrt{\frac{2(R + K)\log 2 + 2\log\frac{1}{\delta}}{N}}\right\}.$$

Here, we used the fact that the observations $f(\mathbf{x})$ are bounded in the domain $[-1, 1]$ and that we can first partition the domain into groups $X_k$ ($K$ subsets) in addition to partitioning the interval $[-1, 1]$ into $R$ smaller sub-intervals and using the Lipschitz constant. Choosing $R = N^{\frac{1}{3}}$ and simplifying gives us with a probability of at least $1 - \delta$:

$$\left|\mathbb{E}_\mathcal{D}[l(\mathbf{h}_\gamma, \mathbf{x})] - \mathbb{E}_\mathcal{S}[l(\mathbf{h}_\gamma, \mathbf{x})]\right| \leq \frac{4(2 + \frac{1}{\gamma})}{N^{\frac{1}{3}}} + 2\sqrt{\frac{2K + 2\log\frac{1}{\delta}}{N}}.$$

Define $\mathbf{h}_\gamma^\star$ to be the minimizer of:

$$(\gamma/2)\mathbb{E}[h(\mathbf{x})^2] - \mathbb{E}[h(\mathbf{x})\cdot f(\mathbf{x})],$$

subject to the fairness constraints. Then, the same generalization bound above also applies to the decision rule $\mathbf{h}_\gamma^\star$ because the $\epsilon$-cover (Definition 1 in [Xu and Mannor, 2012]) is independent of the

choice of the thresholds. By the union bound, we have with a probability of at least $1 - \delta$, *both* of the following inequalities hold:

$$\left|\mathbb{E}_{\mathcal{D}}[l(\mathbf{h}_{\gamma}, \mathbf{x})] - \mathbb{E}_{\mathcal{S}}[l(\mathbf{h}_{\gamma}, \mathbf{x})]\right| \leq \frac{4(2 + \frac{1}{\gamma})}{N^{\frac{1}{3}}} + 2\sqrt{\frac{2K + 2\log\frac{2}{\delta}}{N}} \tag{16}$$

$$\left|\mathbb{E}_{\mathcal{D}}[l(\mathbf{h}_{\gamma}^{\star}, \mathbf{x})] - \mathbb{E}_{\mathcal{S}}[l(\mathbf{h}_{\gamma}^{\star}, \mathbf{x})]\right| \leq \frac{4(2 + \frac{1}{\gamma})}{N^{\frac{1}{3}}} + 2\sqrt{\frac{2K + 2\log\frac{2}{\delta}}{N}}. \tag{17}$$

In particular:

$$\mathbb{E}_{\mathcal{D}}[l(\mathbf{h}_{\gamma}, \mathbf{x})] \leq \mathbb{E}_{\mathcal{S}}[l(\mathbf{h}_{\gamma}, \mathbf{x})] + \frac{4(2 + \frac{1}{\gamma})}{N^{\frac{1}{3}}} + 2\sqrt{\frac{2K + 2\log\frac{2}{\delta}}{N}}$$

$$\leq \mathbb{E}_{\mathcal{S}}[l(\mathbf{h}_{\gamma}^{\star}, \mathbf{x})] + \gamma + \frac{4(2 + \frac{1}{\gamma})}{N^{\frac{1}{3}}} + 2\sqrt{\frac{2K + 2\log\frac{2}{\delta}}{N}}$$

$$\leq \mathbb{E}_{\mathcal{D}}[l(\mathbf{h}_{\gamma}^{\star}, \mathbf{x})] + \gamma + \frac{8(2 + \frac{1}{\gamma})}{N^{\frac{1}{3}}} + 4\sqrt{\frac{2K + 2\log\frac{2}{\delta}}{N}}.$$

The first inequality follows from Equation (16). The second inequality follows from the fact that $\mathbf{h}_{\gamma}$ is an empirical risk minimizer to the regularized loss. The last inequality follows from Equation (17).

Finally, we know that the thresholding rule $\mathbf{h}_{\gamma}^{\star}$ with width $\gamma > 0$ is, by definition, a minimizer to:

$$(\gamma/2)\mathbb{E}[h(\mathbf{x})^2] - \mathbb{E}[h(\mathbf{x}) \cdot f(\mathbf{x})]$$

among all possible bounded functions $h : \mathcal{X} \to [0, 1]$ subject to the desired fairness constraints. Therefore, we have:

$$(\gamma/2)\mathbb{E}[\mathbf{h}_{\gamma}^{\star}(\mathbf{x})^2] - \mathbb{E}[\mathbf{h}_{\gamma}^{\star}(\mathbf{x}) \cdot f(\mathbf{x})] \leq (\gamma/2)\mathbb{E}[\mathbf{h}^{\star}(\mathbf{x})^2] - \mathbb{E}[\mathbf{h}^{\star}(\mathbf{x}) \cdot f(\mathbf{x})]$$

Hence:

$$\mathbb{E}[l(\mathbf{h}_{\gamma}^{\star}, \mathbf{x})] = -\mathbb{E}[\mathbf{h}_{\gamma}^{\star}(\mathbf{x}) \cdot f(\mathbf{x})] \leq \gamma + \mathbb{E}[l(\mathbf{h}^{\star}, \mathbf{x})]$$

This implies the desired bound:

$$\mathbb{E}_{\mathcal{D}}[l(\tilde{\mathbf{h}}_{\gamma}, \mathbf{x})] \leq \mathbb{E}_{\mathcal{D}}[l(\mathbf{h}^{\star}, \mathbf{x})] + 2\gamma + \frac{8(2 + \frac{1}{\gamma})}{N^{\frac{1}{3}}} + 4\sqrt{\frac{2K + 2\log\frac{2}{\delta}}{N}}.$$

Therefore, we have consistency if $N \to \infty$, $\gamma \to 0^+$ and $\gamma N^{\frac{1}{3}} \to \infty$. For example, this holds if $\gamma = O(N^{-\frac{1}{6}})$.

So far, we have assumed that the output of the original classifier coincides with the Bayes regressor. If the original classifier is Bayes consistent, i.e. $\mathbb{E}[|2\eta(\mathbf{x}) - 1 - f(\mathbf{x})|] \to 0$ as $N \to \infty$, then we have Bayes consistency of the post-processing rule by the triangle inequality.

## C   Proof of Proposition 1

*Proof.* Since $|\xi_{\gamma}'(w)| \leq 1$ (see Equation 2), the derivative squared in the stochastic loss in Equation 8 w.r.t. the optimization variable $\gamma$ at a point $\mathbf{x}$ is bounded by $(1 + \rho + \epsilon)^2$ at all rounds. The same holds for the other optimization variable $\mu$. Therefore, the norm squared of the gradient w.r.t. $(\lambda, \mu)$ is bounded by $2(1 + \rho + \epsilon)^2$. Following the proof steps of [Boyd and Mutapcic, 2008] and using the fact that projections are contraction mappings, one obtains:

$$\sum_{t=1}^{T}\left(\mathbb{E}[F^{(t)}] - F^{\star}\right) \leq \frac{||\mu^{\star}||_2^2 + ||\lambda^{\star}||_2^2 + 2(1 + \rho + \epsilon)^2 T\alpha^2}{2\alpha}$$

$$= (1 + \rho + \epsilon)^2 \alpha T + \frac{||\mu^{\star}||_2^2 + ||\lambda^{\star}||_2^2}{2\alpha}.$$

Dividing both sides by $T$, we have by Jensen's inequality $\frac{1}{T}\sum_{t=1}^{T}\mathbb{E}[F^{(t)}] \geq \mathbb{E}[F(\bar{\lambda},\bar{\mu})]$. Plugging this into the earlier results yields:

$$\mathbb{E}[\bar{F}] - F^{\star} \leq (1 + \rho + \epsilon)^2 \alpha + \frac{||\mu^{\star}||_2^2 + ||\lambda^{\star}||_2^2}{2T\alpha}.$$

$\square$

## D  Extension to Other Criteria

### D.1  Controlling the Covariance

The proposed algorithm can be adjusted to control bias according to other criteria as well besides statistical parity. For example, we demonstrate in this section how the proposed post-processing algorithm can be adjusted to control the *covariance* between the classifier's prediction and the sensitive attribute when both are binary random variables.

Let $\mathbf{a}, \mathbf{b}, \mathbf{c} \in \{0, 1\}$ be random variables. Let $C(\mathbf{a}, \mathbf{b}) \doteq \mathbb{E}[\mathbf{a} \cdot \mathbf{b}] - \mathbb{E}[\mathbf{a}] \cdot \mathbb{E}[\mathbf{b}]$ be their covariance, and $C(\mathbf{a}, \mathbf{b} \mid \mathbf{c})$ their covariance conditioned on $\mathbf{c}$:

$$C(\mathbf{a}, \mathbf{b} \mid \mathbf{c} = c) = \mathbb{E}[\mathbf{a} \cdot \mathbf{b} \mid \mathbf{c} = c] - \mathbb{E}[\mathbf{a} \mid \mathbf{c} = c] \cdot \mathbb{E}[\mathbf{b} \mid \mathbf{c} = c]. \tag{18}$$

Then, one possible criterion for measuring bias is to measure the conditional/unconditional covariance between the classifier's predictions and the sensitive attribute when both are binary random variables. Because the random variables are binary, it is straightforward to show that achieving zero covariance implies independence. Hence, this is equivalent to statistical parity when $\epsilon = 0$. The advantage of this formulation, as will be shown next, is that it does not include a hyperparameter $\rho$. The disadvantage, however, is that it can only accommodate binary sensitive attributes.

Suppose we have a binary classifier on the instance space $\mathcal{X}$. We would like to construct an algorithm for post-processing the predictions made by that classifier such that we guarantee $|C(f(\mathbf{x}), 1_S(\mathbf{x}) \mid \mathbf{x} \in X_k)| \leq \epsilon$, where $\mathcal{X} = \cup_k X_k$ is a total partition of the instance space. Informally, this states that the fairness guarantee with respect to the senstiive attribute $1_S : \mathcal{X} \to \{0, 1\}$ holds within each subgroup $X_k$.

We assume, again, that the output of the classifier $f : \mathcal{X} \to [-1, +1]$ is an estimate to $2\eta(x) - 1$, where $\eta(x) = p(\mathbf{y} = 1 | \mathbf{x} = x)$ is the Bayes regressor and consider randomized rules of the form:

$$h : \{0, 1\} \times \{1, 2, \ldots, K\} \times [-1, 1] \to [0, 1],$$

whose arguments are: (i) the sensitive attribute $1_S : \mathcal{X} \to \{0, 1\}$, (ii) the sub-group membership $k : \mathcal{X} \to [K]$, and (iii) the original classifier's score $f(x)$. Because randomization is sometimes necessary as proved in Section 4, $h(x)$ is the probability of predicting the positive class when the instance is $x \in \mathcal{X}$.

Similar to before, if we have a training sample of size $N$, which we will denote by $\mathcal{S}$, we denote $S_k = \mathcal{S} \cap X_k$. The desired fairness constraint on the covariance can be written as:

$$\frac{1}{|S_k|} \Big| \sum_{x_i \in S_k} (1_S(i) - \rho_k) h(x_i) \Big| \leq \epsilon,$$

where $\rho_k = \mathbb{E}_{\mathbf{x}}[1_S(\mathbf{x}) \mid \mathbf{x} \in X_k]$. This is because:

$$\frac{1}{|S_k|} \sum_{x_i \in S_k} (1_S(i) - \rho_k) h(x_i) = \frac{1}{|S_k|} \sum_{x_i \in S_k} 1_S(i) h(x_i) - \frac{\rho_k}{|S_k|} \sum_{x_i \in S_k} h(x_i)$$

$$= \mathbb{E}[1_S(\mathbf{x}) \cdot h(\mathbf{x}) \mid \mathbf{x} \in S_k] - \mathbb{E}[1_S(\mathbf{x}) \mid \mathbf{x} \in S_k] \cdot \mathbb{E}[h(\mathbf{x}) \mid \mathbf{x} \in S_k]$$

$$= C(h(\mathbf{x}), 1_S(\mathbf{x}) \mid \mathbf{x} \in S_k),$$

where the expectation is over the training sample. Therefore, in order to learn $h$, we solve the regularized optimization problem:

$$\min_{0 \leq h(x_i) \leq 1} \quad \sum_{i=1}^{N} (\gamma/2) h(x_i)^2 - f(x_i) h(x_i)$$

$$\text{s.t.} \quad \forall k \in [K] : \Big| \sum_{x_i \in S_k} (1_S(i) - \rho_k) h(x_i) \Big| \leq \epsilon_k \tag{19}$$

where $\gamma > 0$ is a regularization parameter and $\epsilon_k = |S_k|\,\epsilon$. This is of the same general form analyzed in Section A. Hence, the same algorithm can be applied with $b = 0$ and $z_i = 1_S(i) - \rho_k$.

## D.2 Impossibility Result

The previous algorithm for controlling covariance requires that the subgroups $X_k$ be known in advance. Indeed, our next impossibility result shows that this is, in general, necessary. In other words, a deterministic classifier $h : \mathcal{X} \to \{0, 1\}$ cannot be universally unbiased with respect to a sensitive class $S$ across all possible known and unknown groups unless the representation $\mathbf{x}$ has zero mutual information with the sensitive attribute or if $h$ is constant almost everywhere. As a corollary, the groups $X_k$ have to be known *in advance*.

**Proposition 2** (Impossibility result). *Let $\mathcal{X}$ be the instance space and $\mathcal{Y} = \{0, 1\}$ be a target set. Let $1_S : \mathcal{X} \to \{0, 1\}$ be an arbitrary (possibly randomized) binary-valued function on $\mathcal{X}$ and define $\gamma : \mathcal{X} \to [0, 1]$ by $\gamma(x) = p(1_S(\boldsymbol{x}) = 1 \,|\, \boldsymbol{x} = x)$, where the probability is evaluated over the randomness of $1_S : \mathcal{X} \to \{0, 1\}$. Write $\bar{\gamma} = \mathbb{E}_{\boldsymbol{x}}[\gamma(\boldsymbol{x})]$. Then, for any binary predictor $h : \mathcal{X} \to \{0, 1\}$ it holds that*

$$\sup_{\pi:\mathcal{X}\to\{0,1\}} \left\{ \mathbb{E}_{\pi(\boldsymbol{x})} \left| \mathcal{C}\big(h(\boldsymbol{x}), \gamma(\boldsymbol{x})|\,\pi(\boldsymbol{x})\big)\right| \right\} \geq \frac{1}{2}\, \mathbb{E}_{\boldsymbol{x}}|\gamma(\boldsymbol{x}) - \bar{\gamma}| \cdot \min\{\mathbb{E}f, 1 - \mathbb{E}f\}, \qquad (20)$$

*where $\mathcal{C}\big(f(\boldsymbol{x}), \gamma(\boldsymbol{x})|\,\pi(\boldsymbol{x})\big)$ is defined in Equation 18.*

*Proof.* Fix $0 < \beta < 1$ and consider the subset:
$$W = \{x \in \mathcal{X} : \ (\gamma(x) - \bar{\gamma}) \cdot (f(x) - \beta) > 0\},$$

and its complement $\bar{W} = \mathcal{X} \setminus W$. Since $f(x) \in \{0, 1\}$, the sets $W$ and $\bar{W}$ are independent of $\beta$ as long as it remains in the open interval $(0, 1)$. More precisely:

$$W = \begin{cases} \gamma(x) - \bar{\gamma} > 0 & \wedge & f(x) = 1 \\ \gamma(x) - \bar{\gamma} \leq 0 & \wedge & f(x) = 0 \end{cases}$$

Now, for any set $X \subseteq \mathcal{X}$, let $p_X$ be the projection of the probability measure $p(x)$ on the set $X$ (i.e. $p_X(x) = p(x)/p(X)$). Then, with a simple algebraic manipulation, one has the identity:

$$\mathbb{E}_{\mathbf{x}\sim p_X}[(\gamma(\mathbf{x}) - \bar{\gamma})(f(\mathbf{x}) - \beta)] = C(\gamma(\mathbf{x}), f(\mathbf{x}); \mathbf{x} \in X) + (\mathbb{E}_{\mathbf{x}\sim p_X}[\gamma] - \bar{\gamma}) \cdot (\mathbb{E}_{\mathbf{x}\sim p_X}[f] - \beta) \tag{21}$$

By definition of $W$, we have:

$$\mathbb{E}_{\mathbf{x}\sim p_W}[(\gamma(\mathbf{x}) - \bar{\gamma})(f(\mathbf{x}) - \beta)] = \mathbb{E}_{\mathbf{x}\sim p_W}[|\gamma(\mathbf{x}) - \bar{\gamma}||f(\mathbf{x}) - \beta|]$$
$$\geq \min\{\beta, 1 - \beta\}\mathbb{E}_{\mathbf{x}\sim p_W}|\gamma(\mathbf{x}) - \bar{\gamma}|$$

Combining this with Equation (21), we have:

$$C(\gamma(\mathbf{x}), f(\mathbf{x}); \mathbf{x} \in W) \geq \min\{\beta, 1 - \beta\}\mathbb{E}_{\mathbf{x}\sim p_W}|\gamma(\mathbf{x}) - \bar{\gamma}| + (\mathbb{E}_{\mathbf{x}\sim p_W}[\gamma] - \bar{\gamma})(\beta - \mathbb{E}_{\mathbf{x}\sim p_W}[f]) \tag{22}$$

Since the set $W$ does not change when $\beta$ is varied in the open interval $(0, 1)$, the lower bound holds for any value of $\beta \in (0, 1)$. We set:

$$\beta = \bar{f} \doteq \frac{1}{2}\big(\mathbb{E}_{\mathbf{x}\sim p_W} f(\mathbf{x}) + \mathbb{E}_{\mathbf{x}\sim p_{\bar{W}}} f(\mathbf{x})\big) \tag{23}$$

Substituting the last equation into Equation (22) gives the lower bound:

$$C(\gamma(\mathbf{x}), f(\mathbf{x}); \mathbf{x} \in W) \geq$$
$$\min\{\bar{f}, 1 - \bar{f}\} \cdot \mathbb{E}_{\mathbf{x}\sim p_W}|\gamma(\mathbf{x}) - \bar{\gamma}| + \frac{1}{2}(\mathbb{E}_{\mathbf{x}\sim p_W}[\gamma] - \bar{\gamma})\big(\mathbb{E}_{\mathbf{x}\sim p_W} f(\mathbf{x}) - \mathbb{E}_{\mathbf{x}\sim p_{\bar{W}}} f(\mathbf{x})\big) \tag{24}$$

Repeating the same analysis for the subset $\bar{W}$, we arrive at the inequality:

$$C(\gamma(\mathbf{x}), f(\mathbf{x}); \mathbf{x} \in \bar{W})$$
$$\leq -\min\{\bar{f}, 1 - \bar{f}\}\,\mathbb{E}_{\mathbf{x}\sim p_{\bar{W}}}|\gamma(\mathbf{x}) - \bar{\gamma}| + \frac{1}{2}(\mathbb{E}_{\mathbf{x}\sim p_{\bar{W}}}[\gamma] - \bar{\gamma})\big(\mathbb{E}_{\mathbf{x}\sim p_W} f(\mathbf{x}) - \mathbb{E}_{\mathbf{x}\sim p_{\bar{W}}} f(\mathbf{x})\big) \tag{25}$$

Writing $\pi(x) = 1_W(x)$, we have by the reverse triangle inequality:

$$\mathbb{E}_{\pi(\mathbf{x})} \left| \mathcal{C}\big(f(\mathbf{x}), \gamma(\mathbf{x}); \pi(\mathbf{x})\big)\right| \geq \min\{\bar{f}, 1 - \bar{f}\} \cdot \mathbb{E}_{\mathbf{x}} |\gamma(\mathbf{x}) - \bar{\gamma}|. \tag{26}$$

Finally:

$$2\bar{f} \geq p(\mathbf{x} \in W) \cdot \mathbb{E}_{\mathbf{x} \sim p_W} f(\mathbf{x}) \,+\, p(\mathbf{x} \in \bar{W}) \cdot \mathbb{E}_{\mathbf{x} \sim p_{\bar{W}}} f(\mathbf{x}) = \mathbb{E}[f].$$

Similarly, we have $2(1 - \bar{f}) \geq 1 - \mathbb{E}[f]$. Therefore:

$$\min\{\bar{f},\, 1 - \bar{f}\} \geq \frac{1}{2} \min\{\mathbb{E}f,\, 1 - \mathbb{E}f\}.$$

Combining this with Equation (26) establishes the statement of the proposition. $\qquad\square$

## E    Training at Scale Experiment Setup

### E.1    Architectures

The 16 DNN models are:

1. **S-R50x1/1**: A BiT ResNet50 model pretrained on ILSRCV2012.
2. **M-R50x1/1**: A BiT ResNet50 model pretrained on ImageNet-21k.
3. **L-R50x1/1**: A BiT ResNet50 model pretrained on JFT-300M.
4. **S-R50x3/1**: A BiT ResNet50 model 3x wide, pretrained on ILSRCV2012.
5. **M-R50x3/1**: A BiT ResNet50 model 3x wide, pretrained on ImageNet-21k.
6. **L-R50x3/1**: A BiT ResNet50 model 3x wide, pretrained on JFT-300M.
7. **S-R101x1/1**: A BiT ResNet101 model pretrained on ILSRCV2012.
8. **M-R101x1/1**: A BiT ResNet101 model pretrained on ImageNet-21k.
9. **L-R101x1/1**: A BiT ResNet101 model pretrained on JFT-300M.
10. **S-R101x3/1**: A BiT ResNet101 model 3x wide pretrained on ILSRCV2012.
11. **M-R101x3/1**: A BiT ResNet101 model 3x wide pretrained on ImageNet-21k.
12. **L-R101x3/1**: A BiT ResNet101 model 3x wide pretrained on JFT-300M.
13. **MobileNetV2**: pretrained on ILSRCV2012 [Howard et al., 2017].
14. **DenseNet121**: pretrained on ILSRCV2012 [Huang et al., 2017].
15. **DenseNet169**: pretrained on ILSRCV2012 [Huang et al., 2017].
16. **NASNetMobile**: pretrained on ILSRCV2012 [Zoph et al., 2018].

Big Transfer (BiT) models are described in Kolesnikov et al. [2020].

### E.2    Downstream Tasks

The downstream classification tasks are all in CelebA [Liu et al., 2015] and COCO datasets [Lin et al., 2014]. In CelebA, we choose seven attributes that are not immediately related to sex: (1) Smiling, (2) Young, (3) Attractiveness, (4) Narrow Eyes, (5) Oval Face, (6) Pale Skin, and (7) Pointy Nose. We reiterate that we conduct experiments on such vision tasks as a way of validating the technical claims of this paper. Our experiments are not to be interpreted as an endorsement of these visions tasks.

In COCO, we choose the most frequent five objects among images that contain individuals whose sex can be reliably identified from the image caption (see Section 5). The five objects are: Chair (Object ID: 56), Car (Object ID: 2), Handbag (Object ID: 26), Skateboard (Object ID: 36), Tennis Racket (Object ID: 38).