# OpenReview forum: "A Near-Optimal Algorithm for Debiasing Trained Machine Learning Models"
_NeurIPS.cc/2021/Conference — NeurIPS 2021 Poster_

### Official Review · Reviewer_KuGk · 2021-07-11

**Rating:** 7
**Confidence:** 4

**Summary:**

This paper proposes a near-optimal post-processing algorithm (via a regularized optimization objective) that is theoretically guaranteed to debias trained machine learning models for the problem of statistical parity, which is both scalable and performs favorably relative to several existing post-processing and in-processing techniques on 2 datasets from the UCI Machine Learning repo, CelebA, and COCO. The focus of the paper is on binary classification, and the key is to output randomized predictions around the classifier’s threshold.

**Limitations And Societal Impact:**

The authors stated that their results are only applicable to the setting of statistical parity, and made it clear that their method is not intended to address all issues in fair machine learning. They did not discuss potential negative societal impacts of their work, as they are developing an algorithm to mitigate bias in trained machine learning models.

**Main Review:**

Originality:
- The method builds upon the existing work in fair ML, where the proposed approach allows for a randomized thresholding rule that mitigates the downsides of learning a deterministic thresholding rule. This randomization is key for the paper’s nice theoretical properties. The authors adequately cited related work in Section 2 and clearly outlined how their approach differs from existing work. They also provided an overview of baselines they compared to in Section 5, which I appreciated.

Quality:
- The paper is technically sound and claims well-supported.

Clarity:
- The paper was clearly written and easy to follow, barring a few minor points of confusion (listed at the bottom of this text box). Example 1 was a nice illustration of why randomization is necessary, and the notation used in the main text was clear.
- I did have some trouble parsing the experimental results. For example, where exactly are the results for COCO? I saw some in Figures 3-5 but from the main text I expected the results to be included with all the other tables (e.g. results for CelebA, DCCC, etc.). Also how should I be interpreting Figure 2? It’s a bit cramped and the caption doesn’t really tell me what the takeaway is.

Significance:
- The results are significant, as debiasing machine learning models is an important research area, and most existing works do not provide theoretical guarantees. Since this work explores fairness in the statistical parity setting, and mentions that the algorithm generalizes to non-binary attributes, I could see how others would build upon these ideas.

------
UPDATE: I have read the other reviews as well as the authors' response. Thanks to the authors for clearing up some points of confusion, and I'll keep my score as is.

**Time Spent Reviewing:**

3

---

> ### Author Response · Authors · 2021-08-09
> **Response to the points raised**
>
> Thank you for the detailed review. Please find our response below to the raised points.
>
> *Q: from the main text I expected the [COCO] results to be included with all the other tables.*
>
> A: COCO is used in studying the impact of training at scale so it is used in Figures 3-5 as you mentioned. We will clarify this to avoid confusion.
>
> *Q: Also how should I be interpreting Figure 2?*
>
> A: Figure 2(LEFT) shows the tradeoff curves between bias and accuracy using both the proposed algorithm and Hardt et al., 2016. As mentioned in the caption, the x-axis is the level of bias enforced ($\epsilon$) while the y-axis is the resulting accuracy (higher is better). The objective is to show that the proposed algorithm continues to perform better even when approximate fairness is used. Figure 2(RIGHT) shows the post-processing rules used by both algorithms on CelebA in ResNet50 for illustration purposes only.

---

### Official Review · Reviewer_Apbv · 2021-07-16

**Rating:** 7
**Confidence:** 3

**Summary:**

In this paper, the authors propose a post-processing method for debiasing trained models. The authors defined statistical parity as a 'bias'. The paper proposes a constrained optimization problem that takes the input data, sensitive attribute, partitioning and a trained model to yield a probabilistic decision rule. The authors proved theoretical properties and empirically validate its advantages on standard benchmark datasets across both classical algorithms.

**Limitations And Societal Impact:**

The authors did not address the limitations and potential societal impact of their work in the paper.

**Main Review:**

Fairness is an essential topic in the machine learning community. The proposed method suggests a post-processing algorithm for reducing bias from trained models according to statistical parity. Although 'bias' is a societal concept that cannot be adequately measured by statistical parity, the reviewer thinks the proposed method suggests a reasonably acceptable solution.

For a minor comment, the reviewer thinks some explanations are relatively complicated (for instance, Example 1) due to the page limit. Some revision would be necessary for readability.

**Conclusion**
The reviewer thinks the paper is well-written and generally acceptable, which will be an excellent contribution to the NeurIPS conference.

**Time Spent Reviewing:**

6

---

> ### Author Response · Authors · 2021-08-09
> **Response**
>
> Thank you for the detailed review, please find our response below.
>
> *Q: Some explanations are relatively complicated (for instance, Example 1) due to the page limit.*
>
> A: The objective of Example 1 is to prove that randomization at the thresholds is, in general, necessary for Bayes risk consistency. We show that randomization is of practical concern when using models that produce discrete scores similar to the model discussed in Example 1, such as kNN and in deep neural networks. The example is complementary to the main theorem and we will try to provide more context which would hopefully make it less involved.

---

### Official Review · Reviewer_EmDZ · 2021-07-19

**Rating:** 5
**Confidence:** 4

**Summary:**

The paper introduces a post-processing method to enforce demographic parity. The authors provide theoretical guarantees for their approach as well as empirically evaluate its performance on various datasets and against several baselines.

**Limitations And Societal Impact:**

There is not much discussion on the limitation of the work. On the societal impact question 1(c), the authors answer: "We propose a debiasing algorithm for machine learning. We do not foresee any potential negative societal impacts of our work." I think at the least there should be references to the literature that highlights where demographic parity is not a reasonable notion of fairness. Moreover, I find the task of predicting the attractiveness of people from their images a bit inappropriate at the least. While the authors do not endorse using AI for such tasks, it would have been much better if other datasets are used (e.g., there are datasets for facial recognition algorithms where the prediction bias happens because of the skin color).

**Main Review:**

Post-processing fairness-enforcing approaches are convenient as they can interact with black-box models as well as enjoy faster running times compared to in-processing methods. The paper is a nice addition to the growing literature on post-processing. It is known that post-processing can be suboptimal so it is important to understand the limits of such approaches.

However, I do have some concerns about the paper:

--The bound on Theorem 2 depends on both $\gamma$ and $1/\gamma$. How should this parameter be selected in practice?  In particular, how should the bound in Theorem 2 be interpreted as constant $\gamma$ is used in the experiments? Why a fresh set of data points are needed in the statement of Theorem 2? Moreover, for the theory results, it would be great to add a discussion of the applicability and limitations. When should your approach be used? Are there any properties in the dataset or the black-box that make your approach more suitable compared to prior work?

--Some of the details in the experiments are not clear and some justifications are needed. I have not checked the appendix to see whether these details are provided. For example, why additional bias is introduced in DCCC? Why the performance is not reported for the reduction-based approach on CelebA? The details of averaging and normalization for the results in the scaling section are not described clearly. More particularly, I am curious about the following points:

        --How does the running time compare to Hardt et. al.? Both approaches seem to achieve the same level of fairness violation while the approach of Hardt et al is much simpler.

        --Why is there no comparison with Wei et. al, 2020 as they are also proposing a post-processing approach?

        --Imposing post-processing seems to improve accuracy for DCCC. This is confusing. As mentioned above, why additional bias is introduced in DCCC, and how does the accuracy is measured? i.e. for the original data or the biased data?

        --The results in Table 2 for DCCC are a bit confusing. Why does the error get larger as more fairness violation is allowed? Also, there seems to be no real trade-off in this dataset. So what is the takeaway?

        --I cannot follow the results in the scaling section. The details of the averaging and normalization are missing. There is no mention of the bias in this section and all the reported performances are based on normalized error. What is the takeaway here? Why there is no comparison with baselines?

--The choice of using CelebA dataset to predict attractiveness as a question regarding fairness is not appropriate. Please see my comments below.

Minor comments and questions:

--I would prefer to call this fairness notion demographic parity. Statistical parity refers to a more broad category of fairness metrics where parity on some statistical notion is enforced for different subgroups.

--Can the result in Example 1 be extended to approximate statistical parity instead of exact? I am curious whether randomization is truly necessary there.

--What is $n$ in Theorem 1? Is it supposed to be $N$?

This is a truly borderline paper. It can get accepted or rejected with an equal chance though I am leaning a bit towards rejection at the moment.

**Time Spent Reviewing:**

6 hours

---

> ### Author Response · Authors · 2021-08-09
> **Response to the points raised**
>
> Thank you for the detailed review. Please find our response below to the raised points.
>
> *Q: The bound in Theorem 2 depends on both $\gamma$ and $1/\gamma$.*
>
> A: The dependence of the bound on both $\gamma$ and $1/\gamma$ is typical in the literature (see for instance Corollary 13.8 in [1] and Theorem 7 in [2]); otherwise the optimal value of the hyperparameter would be either 0 or $\infty$. One way to choose $\gamma$ is to minimize the upper bound, as we do in the proof of Theorem 2, or based on a separate validation dataset (Line 213).
>
> *Q: Why a fresh set of data points are needed in the statement of Theorem 2?*
>
> A: These are required not just in theory, but also in practice to ensure that the scores have the same distribution as future test data. The scores of the original training examples have a different distribution because the model is trained on them (e.g. they would be concentrated around the extremes when using deep neural nets).
>
> *Q: Are there any properties in the dataset or the black-box that make your approach more suitable compared to prior work?*
>
> A: We discuss this in the Related Work section as well as in Experiments. The postprocessing algorithm learns to randomize around the threshold and this is important both in theory and practice; e.g. when the model’s scores are concentrated in a few points such as in kNN or deep neural networks. We demonstrate the need for randomization in such cases in Example 1 and experimentally in Table 1.
>
> *Q: Why is additional bias introduced in DCCC? Imposing post-processing seems to improve accuracy for DCCC. This is confusing.*
>
> A: We introduced bias in DCCC to study the case in which bias shows up in the training data only (e.g. due to the data curation process) but the test data remains unbiased (see for example [3] and [4] which discuss similar situations in common benchmark datasets such as ImageNet). Because test data is unbiased, debiasing models can improve accuracy because it acts as a regularizer in which a condition known to hold at test time is enforced during training. This is why accuracy improves for DCCC.
>
> *Q: Why is the performance not reported for the reduction-based approach on CelebA?*
>
> A: Because the reduction approach requires retraining the deep neural network several times in sequence before convergence, which is quite expensive for deep models such as ResNet50 and MobileNet.
>
> *Q: Both approaches seem to achieve the same level of fairness violation while the approach of Hardt et al is much simpler.*
>
> A: Both approaches achieve the same level of fairness but the impact on accuracy is quite different. The proposed algorithm maintains significantly higher accuracy.
>
> *Q: I cannot follow the results in the scaling section.*
>
> A: We describe the normalization in Lines 271-273. Here, “normalized error” is a soft ranking of models. In each downstream task, the best model is assigned a rank of 0 while the worst model is assigned a rank of 1. All other models are assigned ranks in between linearly using: (model_error - min_error)/(max_error - min_error). This is carried out for each downstream task separately. So, we have different rankings of models for different downstream tasks. After that, we report the average ranking. We will further clarify this in the revised paper. The main takeaway is that the proposed postprocessing algorithm performs better for models trained at scale (by either scaling up the model size or the dataset size), which is quite encouraging given that the recent progress is driven primarily by scaling-based approaches.
>
> *Q: The choice of using CelebA dataset to predict attractiveness as a question regarding fairness is not appropriate.*
>
> A: As we explicitly mention in the paper (Lines 51-57, Lines 235-236, as well as in Appendix E.2), we do not endorse predicting facial attributes. However, our goal is to validate claims that are purely technical; namely that the proposed algorithm debiases models without significantly impacting their test accuracy. We use “attractiveness” because it exhibits strong gender-related bias in the CelebA dataset. If the reviewer finds that this is not sufficiently clear, we can provide further clarification in the manuscript.
>
> *Q: I think at the least there should be references to the literature that highlights where demographic parity is not a reasonable notion of fairness.*
>
> A: We briefly mention this in Lines 17-22, where we argue that the definition of bias is application-specific.
>
> *Q: Can the result in Example 1 be extended to approximate statistical parity instead of exact?*
>
> A: Yes, in Example 1, for $\epsilon<12/70$, the optimal rule is to randomize at the threshold x=0. The amount of randomization depends on $\epsilon$. When $\epsilon > 12/70$, the optimal rule does not require randomization.
>
> [1] Shalev-Shwartz, Shai, and Shai Ben-David. Understanding machine learning: From theory to algorithms. Cambridge university press, 2014.
>
> [2] Shalev-Shwartz, Shai, et al. "Stochastic Convex Optimization." COLT, 2009.
>
> [3] Torralba, Antonio, and Alexei A. Efros. "Unbiased look at dataset bias." CVPR 2011. IEEE, 2011.
>
> [4] DeVries, Terrance, et al. "Does object recognition work for everyone?" arXiv:1906.02659 [cs.CV], 2019.

---

### Decision · Program_Chairs · 2021-09-27

**Decision:**

Accept (Poster)

**Comment:**

The balance of the reviews is in favor of accepting the paper. I want to follow this consensus. While some of the reviewers described significant issues with the paper, I do not view these issues as disqualifying. I expect the camera-ready version of the paper will be able to address most of them as per the author response.